# Tick extracellular vesicles enable arthropod feeding and promote distinct outcomes of bacterial infection

Adela S. Oliva Chávez [1,18], Xiaowei Wang[1], Liron Marnin[1], Nathan K. Archer [2], Holly L. Hammond[1], Erin E. McClure Carroll[1,19], Dana K. Shaw [1,20], Brenden G. Tully[3], Amanda D. Buskirk[4,21], Shelby L. Ford[5], L. Rainer Butler[1], Preeti Shahi[1], Kateryna Morozova[6], Cristina C. Clement[6,22], Lauren Lawres[7], Anya J. O' Neal [1], Choukri Ben Mamoun [7], Kathleen L. Mason[8], Brandi E. Hobbs [1], Glen A. Scoles[8,23], Eileen M. Barry [4], Daniel E. Sonenshine[9,10], Utpal Pal[11], Jesus G. Valenzuela[9], Marcelo B. Sztein[4,12,13], Marcela F. Pasetti[4,12], Michael L. Levin[5], Michail Kotsyfakis [14], Steven M. Jay [15], Jason F. Huntley[3], Lloyd S. Miller[2,16], Laura Santambrogio [17,22] & Joao H. F. Pedra [1✉]

Extracellular vesicles are thought to facilitate pathogen transmission from arthropods to humans and other animals. Here, we reveal that pathogen spreading from arthropods to the mammalian host is multifaceted. Extracellular vesicles from *Ixodes scapularis* enable tick feeding and promote infection of the mildly virulent rickettsial agent *Anaplasma phagocytophilum* through the SNARE proteins Vamp33 and Synaptobrevin 2 and dendritic epidermal T cells. However, extracellular vesicles from the tick *Dermacentor andersoni* mitigate microbial spreading caused by the lethal pathogen *Francisella tularensis*. Collectively, we establish that tick extracellular vesicles foster distinct outcomes of bacterial infection and assist in vector feeding by acting on skin immunity. Thus, the biology of arthropods should be taken into consideration when developing strategies to control vector-borne diseases.

[1] Department of Microbiology and Immunology, University of Maryland School of Medicine, Baltimore, MD, USA. [2] Department of Dermatology, Johns Hopkins University School of Medicine, Baltimore, MD, USA. [3] Department of Medical Microbiology and Immunology, University of Toledo College of Medicine and Life Sciences, Toledo, OH, USA. [4] Center for Vaccine Development and Global Health, University of Maryland School of Medicine, Baltimore, MD, USA. [5] Centers for Disease Control and Prevention, Atlanta, GA, USA. [6] Department of Pathology, Albert Einstein College of Medicine, Bronx, NY, USA. [7] Section of Infectious Diseases, Department of Internal Medicine, Yale University School of Medicine, New Haven, CT, USA. [8] USDA, ARS, Animal Disease Research Unit, Washington State University, Pullman, WA, USA. [9] Vector Molecular Biology Section, Laboratory of Malaria and Vector Research, National Institute of Allergy and Infectious Diseases, National Institutes of Health, Rockville, MD, USA. [10] Department of Biological Sciences, Old Dominion University, Norfolk, VA, USA. [11] Department of Veterinary Medicine, University of Maryland, College Park, MD, USA. [12] Department of Pediatrics, University of Maryland School of Medicine, Baltimore, MD, USA. [13] Department of Medicine, University of Maryland School of Medicine, Baltimore, MD, USA. [14] Institute of Parasitology, Biology Center of the Czech Academy of Sciences, Ceske Budejovice, Czech Republic. [15] Fischell Department of Bioengineering, University of Maryland, College Park, MD, USA. [16] Immunology, Janssen Research and Development, Spring House, PA, USA. [17] Department of Microbiology and Immunology, Albert Einstein College of Medicine, Bronx, NY, USA. [18] Present address: Department of Entomology, Texas A&M University, College Station, TX, USA. [19] Present address: Excerpta Medica, Doylestown, PA, USA. [20] Present address: Department of Veterinary Microbiology and Pathology, Washington State University, Pullman, WA, USA. [21] Present address: Center for Drug Evaluation and Research, Office of Pharmaceutical Quality, Office of Process and Facilities, Division of Microbiology Assessment, Microbiology Assessment Branch III, U.S. Food and Drug Administration, Silver Spring, MD, USA. [22] Present address: Department of Radiation Oncology and Physiology and Biophysics, Englander Institute for Precision Medicine, Weill Cornell Medicine, New York, NY, USA. [23] Present address: USDA, ARS, Invasive Insect Biocontrol and Behavior Laboratory, Beltsville, MD, USA. ✉email: jpedra@som.umaryland.edu

Arthropod vectors transmit a diverse portfolio of bacterial, viral, and parasitic agents to mammals[1]. This evolutionary feat is partly due to the secretion of salivary molecules that disrupt host homeostasis and alter inflammation upon blood-feeding[2–4]. Salivary molecules from arthropod vectors redirect host immunity, enable microbial spreading and influence disease outcomes in humans and other animals[2–4]. For instance, saliva from the sandfly *Lutzomyia longipalpis* enhances transmission of the parasite *Leishmania major*[5]. Salivary effectors present in mosquitoes potentiate arbovirus transmission and augment disease severity in the mammalian host[6,7]. Tick molecules inhibit inflammation and increase transmission of viruses and bacteria to a variety of animals[8–11].

Although saliva-assisted transmission appears to be universal among arthropods that feed on blood[2,3,12], the mechanisms that affect interspecific responses reflect unique interactions occurring between vectors and their hosts. As an example, the inflammatory response caused by the bite of the mosquito *Aedes aegypti*, which is classically assumed to defend the mammalian host against arboviral infection, results in higher Semliki forest and Bunyamwera viral load[13]. Depletion of neutrophils and inhibition of inflammasome activity dampens the mosquito's ability to enable viral transmission[13]. Infiltration of neutrophils at the sand fly bite site, which is typically judged to be microbicidal, is important for *L. major* infection and establishment of illness[14]. Altogether, these studies suggest that the relationship between salivary components, host immunity and microbial transmission are complex and ill-defined.

Extracellular vesicles (EVs) are a heterogenous population of nanovesicles that mediate interspecies communication[15–17]. Exosomes, a subset of EVs that range from approximately 50 to 150 nm in size, are enriched for certain proteins, lipids, nucleic acids, and glycoconjugates[15–17]. Recently, exosomes were shown to facilitate arbovirus transmission to mammals[18–21] and two studies examined the molecular cargo of extracellular vesicles originated from the evolutionarily distant Asian longhorned tick *Haemaphysalis longicornis*[22,23]. The functional plasticity and immunomodulatory potential of arthropod exosomes during vector transmission remain mostly unknown. To address this question, we used ticks as arthropod models due to their long-feeding behavior and medical importance[24]. Herein, we adapted a protocol to culture tick salivary glands[25–27] and developed a salivary organoid system that mimics EV release. We also characterized the proteomic composition of these EVs in adult *Ixodes scapularis* and show that EVs derived from cultured cells and salivary glands have a different cargo and post-translational profile.

In this work, we expand the current model for arthropod vector feeding on mammals by showing that tick EVs affect dendritic epidermal T cells (DETCs)[28–30]. DETCs are a specialized subset of γδ T cells involved in wound healing and present exclusively in the murine skin epidermis[28–35]. Furthermore, we indicate that EVs enable tick feeding and foster distinct outcomes of bacterial infection in mammals. EVs from the deer tick *I. scapularis* facilitate infection of the mildly virulent rickettsial agent *Anaplasma phagocytophilum* in the mammalian host. Conversely, EVs from the tick *Dermacentor andersoni* reduce spreading of the deadly pathogen *Francisella tularensis* in a murine model.

## Results

**Ticks secrete salivary effectors within diverse EVs.** We developed a methodology for recovering tick EVs from cells and organs in an EV-depleted medium (Supplementary Fig. 1a, b). EVs from the blacklegged tick *I. scapularis* were heterogenous, as shown by transmission electron microscopy (Fig. 1a). In silico reconstruction of the EV pathway based on the *I. scapularis* genome[36] identified genes associated with exosome biogenesis and secretion (Supplementary Data 1). These molecules included the tetraspanin CD63 and two proteins associated with the endosomal sorting complex required for transport (ESCRT) machinery: (i) α-1,3/1,6-mannosyltransferase interacting protein X (ALIX) and the (ii) tumor susceptibility gene 101 protein (TSG101)[15–17]. Mammalian polyclonal antibodies cross-reacted with markers present in EVs originated from three tick cell lines: (i) *Amblyomma americanum* (i.e., vector of ehrlichiosis[37] – AAE2 cells); (ii) *I. scapularis* (i.e., vector of Lyme disease and Anaplasmosis[24] – ISE6 cells); and (iii) *D. andersoni* (i.e., vector of tularemia[38] – DAE100 cells) (Fig. 1b and Supplementary Fig. 1c–e). These EVs presented an average mean size of $173 \pm 7$, $137 \pm 6$, and $183 \pm 3$ nm, respectively (Supplementary Fig. 1f–h and Supplementary Movies 1–3).

To study the effect of *I. scapularis* EVs, we silenced the expression of two tick soluble N-ethylmaleimide-sensitive factor attachment receptor (SNARE) genes (e.g., *synaptobrevin 2* and *vamp33*) through RNA interference (RNAi). Tick studies rely on delivering small interfering RNA (siRNA) to modulate gene expression because complete genetic ablation within ticks is not currently feasible[39]. No differences in cell viability were observed between the scrambled (sc) control and silenced (si) treatments (Supplementary Fig. 2a). Nanoparticle tracking analysis (NTA) revealed decreased secretion of tick vesicles in the classical exosomal range (i.e., 50–150 nm) for both *synaptobrevin 2* and *vamp33* siRNA-treated cells (Fig. 1c, Supplementary Fig. 2b–d, and Supplementary Movies 4–7).

EVs derived from adult *I. scapularis* salivary glands had an average mean size of $198 \pm 4$ nm (Fig. 1d and Supplementary Movie 8). We then obtained EVs from dissected *I. scapularis* salivary glands cultured ex vivo in a vesicle-free medium to analyze their protein content. High-throughput proteomics analysis of their cargo demonstrated a heterogeneous population with proteins typically featured for microvesicles, exosomes, and exomeres[40] (Supplementary Fig. 2e, f, Supplementary Data 2–5, and data available via ProteomeXchange, identifier PXD018779). Tick proteins that have an impact on the immune response (e.g., sialostatins)[9] and molecules associated with oxidative stress, metabolism, and cell biology processes were overrepresented (Supplementary Figs. 3 and 4 and Supplementary Data 2). The presence of the exosomal markers CD63, ALIX, and TSG101 was detected in tick salivary EVs by immunoblots (Fig. 1e–g), as shown previously[41]. Moreover, a subset of proteins detected in our proteomics analysis was validated through western blots (Fig. 1h). Some proteins were present in all EVs surveyed (e.g., DHX16 and PLS3), while others were only detected in EVs originated from tick cells (e.g., CTNNB1) when compared to salivary glands (e.g., CCT7 and GFPT1). A similar pattern emerged for glycosylation in tick EVs (Fig. 1i), but this observation was less pronounced for phosphorylation (Fig. 1j) and carbonylation (Fig. 1k), a common product of protein oxidation[42].

It was not possible to corroborate all our findings with EVs from tick saliva due to the limited material. Due to this technical limitation, we adopted a hybrid isolation model where we substantiated our results with EVs from tick saliva in *I. ricinus*, the main vector of Lyme disease in Europe[43]. This methodology was also recently used in EVs extracted from *I. scapularis* and *Amblyomma maculatum* to determine their effect on a human keratinocyte cell line[41]. Using this approach, we identified the tick protein Sialostatin L2 (SL2)[9] and the tetraspanin CD63 in EVs derived from adult saliva (Fig. 1l). Altogether, our data suggested that ticks secrete a diverse group of EVs containing molecules capable of modulating mammalian host physiology.

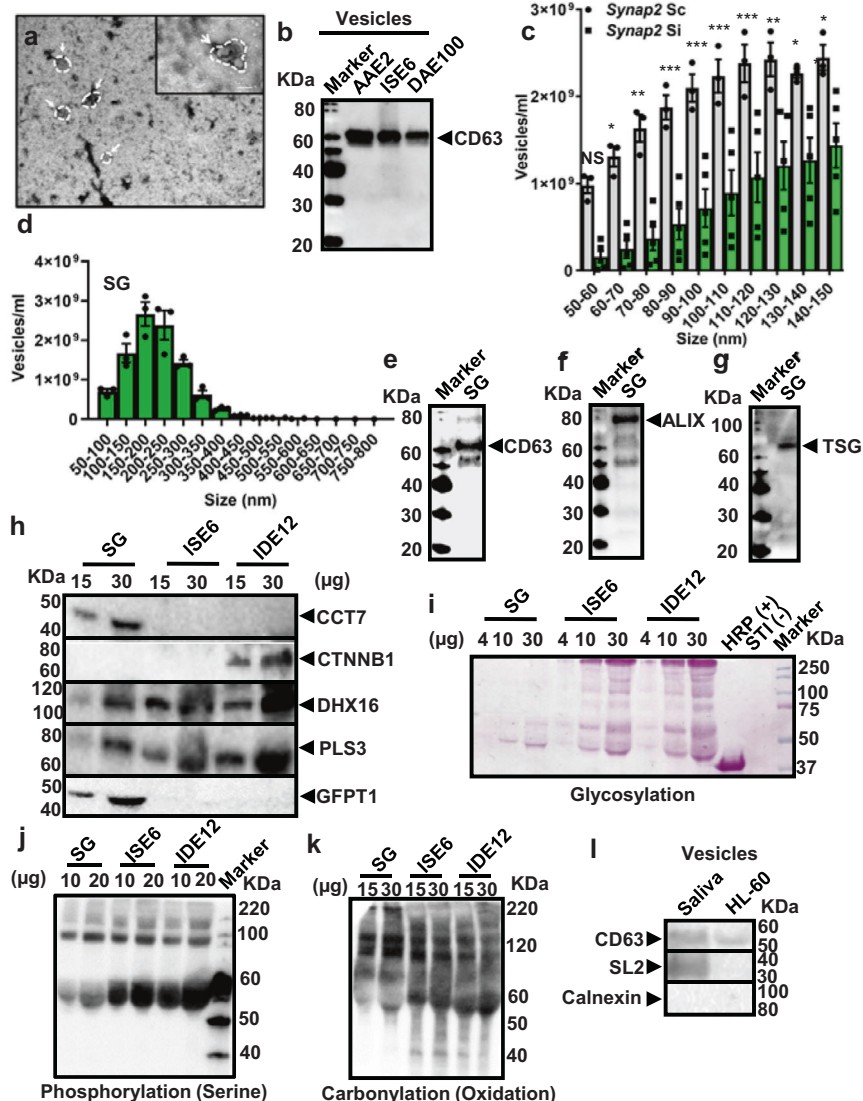

**Fig. 1 Ticks secrete EVs with a distinct cargo profile. a** Transmission electron microscopy of EVs released by *I. scapularis* (ISE6) cells. Scale bar: 100 nm, large panel = 11,000x; small panel = 30,000x. The image is representative of two independent experiments. **b** SDS-PAGE immunoblot showing CD63+ EVs purified from AAE2, ISE6, and DAE100 tick cells (70 μg of protein). Images are a representation of three independent experiments. **c** *I. scapularis* ISE6 cells transfected with *synaptobrevin 2* small interfering (*Synaptobrevin 2 si*) (green) or scrambled control RNA (*Synaptobrevin 2 sc*) (gray) to reduce the EV release. Concentration and size distribution of EVs were measured by nanoparticle tracking analysis (NTA). Mean ± standard error of the mean (SEM) of technical replicates plotted from one of two independent experiments are shown. Similar results were obtained in both independent studies. Two-way ANOVA using size and treatment as variables followed by Sidak's multiple comparison statistical test of the same size vesicles between treatments. *$p < 0.05$; **$p < 0.005$; ***; $p < 0.001$; NS not significant. **d** EVs originated from *I. scapularis* salivary gland (SG) cultures. Mean ± SEM are plotted. Data is representative of three independent experiments. **e–g** SDS-PAGE immunoblots showing CD63+, ALIX+, and TSG101+ EVs purified from *I. scapularis* SGs (24 μg of protein). **h** CCT7, CTNNB1, DHX16, PLS3, and GFPT1 expression in EVs purified from partially engorged *I. scapularis* SG and uninfected tick cells. **i** Glycosylation of proteins in EVs purified from partially engorged *I. scapularis* SG and tick cells. HRP (+) represents the horseradish peroxidase positive control and STI (−) indicates the soybean trypsin inhibitor negative control for the assay. **j, k** Phosphorylation and carbonylation of proteins in EVs purified from partially engorged *I. scapularis* SG and tick cells. **l** EVs purified from *I. ricinus* saliva and immunoblotted against CD63, Sialostatin L2 (SL2), and calnexin (15 μg of protein). EVs from HL-60 were used as control cells, whereas calnexin was used as a technical negative control. Blots are representative of two independent experiments. Source data are provided as a Source Data file.

**Tick EVs interact with mammalian immune cells**. We then postulated that molecules present in EVs bind to immune cells. Proteomics of *I. scapularis* EVs derived from salivary glands revealed an overrepresentation of proteins connected to cell biology, oxidative stress, and metabolism (Fig. 2a, b, Supplementary Figs. 3 and 4, and Supplementary Data 2 and 5). Integrins direct EV tropism to specific organs through molecular interactions in the extracellular matrix[44]. To test this concept, we incorporated a fluorescent dye with long aliphatic tails (PKH26)[45]

into lipid regions of membranes from tick EVs. We also labeled the plasma membrane of murine bone marrow-derived macrophages (BMDMs) pre-treated with cytochalasin D, an inhibitor of actin polymerization, to block phagocytosis. Laser scanning confocal microscopy coupled to live-cell imaging showed PKH26-labeled EVs bound to murine BMDMs (Fig. 2c and Supplementary Movie 9) and time-lapse microscopy displayed saturation at ~160 min (Fig. 2d). Next, we labeled *I. scapularis* EVs with the lipophilic 3,3′-dioctadecyloxacarbocyanine (DiO) dye and murine

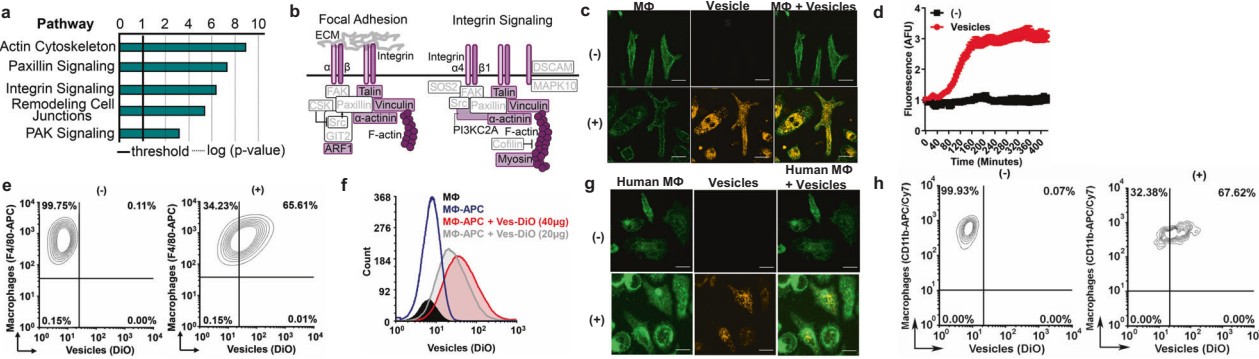

**Fig. 2 Tick EVs bind to mammalian immune cells. a** Overrepresentation of proteins involved in cell adhesion signaling networks in *I. scapularis* salivary gland EVs. The biological relationship between proteins was determined using the right-tailed Fisher's exact test with Benjamini–Hochberg multiple-testing correction. $p < 0.05$; −log ($p$ value) >1.5. **b** Graphic representation of focal adhesion and integrin signaling proteins (purple) found in *I. scapularis* salivary gland EVs. Confocal images of **c** murine bone marrow-derived macrophages (BMDMs) (bar = 20 μm) and **g** macrophages derived from human peripheral blood mononuclear cells (green) bound to *I. scapularis* EVs labeled with PKH26 (orange) (bar = 20 μm), respectively. Cells were pre-treated with cytochalasin D (5 μM). (−) No EVs; (+) EVs. Images are representative of two independent experiments. **d** Arbitrary fluorescent units (AFU) of murine BMDMs bound to PKH26-labeled tick EVs (red). BMDMs incubated in the absence of EVs (black). Each time point represents the mean fluorescence from seven different cells minus background and normalized to time 0. Graphs show one of two independent experiments. Flow cytometry with **e** murine BMDMs (F4/80+-APC) and **h** human macrophages (CD11b+-APC/Cy7) bound to DiO-labeled tick EVs (40 μg), respectively. No EVs (−); EVs (+). Figures are representative of three biological replicates. **f** Flow cytometry analysis of unlabeled BMDMs (black), F4/80-APC-labeled BMDMs (blue), F4/80-APC-labeled BMDMs bound to 40 μg DiO-labeled tick EVs (red), and F4/80-APC-labeled BMDMs bound to 20 μg of DiO-labeled and 20 μg unlabeled tick EVs (gray). The histogram shows one of three biological replicates. PKH26 is a red fluorescent dye with long aliphatic tails[45]. In **c**, **g**, PKH26 was artificially transformed to an orange color to be visualized by color-blind readers. Source data are provided as a Source Data file.

BMDMs with the marker F4/80 conjugated to allophycocyanin (APC). F4/80+ murine macrophages bound to tick EVs in a dose-dependent manner (Fig. 2e, f). Similarly, tick EVs bound to CD11b+ human macrophages originated from peripheral blood mononuclear cells (PBMCs) (Fig. 2g, h). Collectively, our results indicated that arthropod EVs interact with immune cells.

**Tick EVs redirect skin immunity**. To assess whether tick EVs regulate an immune environment in vivo, we evaluated skin cell populations during *I. scapularis* hematophagy. Under normal tick-feeding, we observed increased inflammation in the murine skin, as inferred by an elevated presence of granulocytic and phagocytic cells and greater epidermal thickness (Supplementary Fig. 5). These findings were consistent with tick bites eliciting a visible innate immune response in the skin of humans[46]. Next, we performed RNAi silencing of *vamp33* and *synaptobrevin 2* in *I. scapularis* in vivo. For the sake of simplicity, we denominated arthropods that had *vamp33* or *synaptobrevin 2* gene expression reduced (*vamp33* or *synap2 si*) as EV-deficient ticks. No difference in tick attachment was observed in the silenced versus the control treatment (Fig. 3a and Supplementary Fig. 6a, b). However, diminished feeding was measured for EV-deficient ticks (Fig. 3b and Supplementary Fig. 6c). Impaired tick feeding was linked to a change in the cytokine and chemokine milieu in the mouse skin. Notably, the skin immune environment was skewed towards an adaptive T helper 2 (Th2) wound healing response in mice fed on by EV-deficient ticks (Fig. 3c–e and Supplementary Fig. 7). Surprisingly, there was a 54% increase in the number of gamma delta (γδ) T cells when EV-deficient ticks were placed on mice and compared to the control treatment (Fig. 3f and Supplementary Fig. 8). Increased numbers of γδ T cells at the skin site correlated to favored chemotactic conditions for immune cell recruitment during a tick bite[47] (Fig. 3g–j and Supplementary Fig. 7).

**Depletion of DETCs rescues EV-dependent tick feeding at the skin site**. γδ T cells were the only immune cells that showed an increased at the skin site after impairment of tick EVs (Fig. 3f and

Supplementary Fig. 8). Thus, we determined the effect of skin γδ T cells on tick feeding. We first reduced the number of γδ T cells in the murine skin through antibody depletion (Fig. 4a). We placed ticks injected with *vamp33* siRNA on mice and measured their weight at day 3 post-attachment. We detected a 24% reduction in the weight of EV-deficient ticks for the isotype control treatment (Fig. 4b, weight, two left-most rectangular bars [Isotype Ab]). This decrease in weight was not observed when EV-deficient ticks (*vamp33 si* treatment) were placed on antibody-depleted γδ T cell mice (Fig. 4b, weight, two right-most rectangular bars [Anti-γδ Ab]). Similar results were obtained in animals genetically ablated for γδ T cells (Fig. 4c, weight, comparison in the two left-wildtype [WT] and right-most rectangular bars [*tcrδ−/−*] mice). Importantly, αβ T cells did not play any role in arthropod feeding. The weight of EV-deficient ticks (*vamp33 si* treatment) placed on animals genetically ablated for αβ T cells was reduced when compared to the control treatment (Fig. 4c, weight, comparison in the middle [*tcrβ−/−*]). Taken together, our findings suggested that EVs regulate skin immunity through γδ T cells, providing an advantageous environment for tick feeding.

We then determined whether the effect of tick EVs was on resident or infiltrating γδ T cells. We placed ticks injected with *vamp33* siRNA on mice with or without administering the molecule FTY720. FTY720 inhibits lymphocyte egress from lymph nodes[48] (Fig. 4d). We did not observe any differences in the number of γδ T cells or the weight of EV-deficient ticks placed on FTY720-treated mice when compared to the control treatment (Fig. 4e). These findings indicated that tick EVs affect the resident, but not the infiltrating γδ T cells during feeding.

Tick saliva may be injected in the epidermis, dermis, or the epidermis/dermis border depending on the tick mouthpart apparatus[4,49]. The γδ T cell population in the mouse skin is constituted of two distinct subpopulations, including dermal γδ T cells (also known as γδ T17 cells) that express the T cell receptor Vγ4 or Vγ6[30,50], and DETCs that express the T cell receptor Vγ5Vδ1 in the epidermis[28], according to the Heilig and Tonegawa nomenclature[51]. Thus far the effect of the tick bite has been predominantly reported to the dermis[4,49]. Therefore, we first

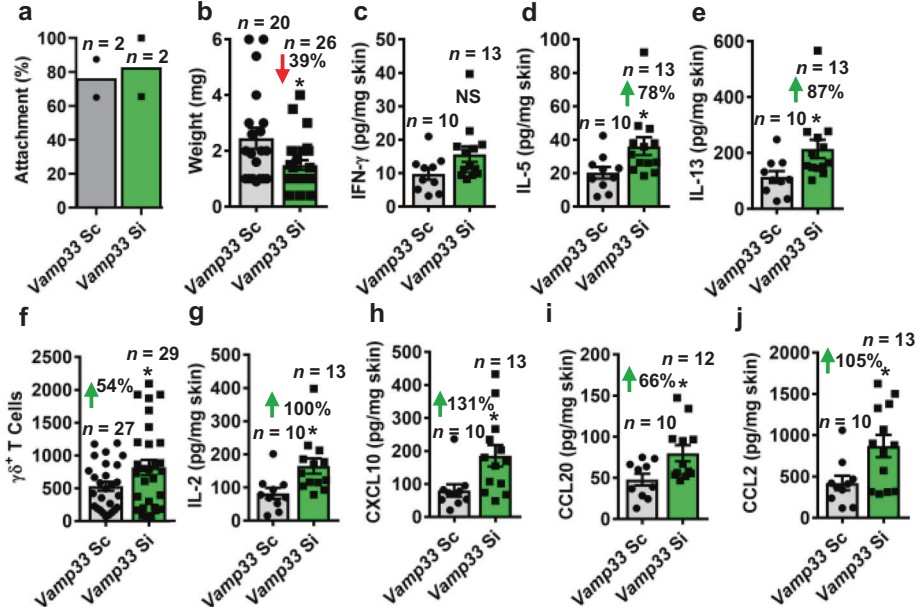

**Fig. 3 Tick EVs affect the skin resident immune environment. a** Attachment and **b** weight of *I. scapularis* ticks microinjected with *vamp33* small interfering (*Vamp33 si*) (green) (n = 26) or scrambled (*Vamp33 sc*) (gray) (n = 20) RNA on C57BL/6 mice (n = 2 each treatment) after 3 days of feeding. Mean ± standard error of the mean (SEM) are plotted. Data represent two independent experiments. Statistical differences were evaluated by two-tailed *t* test. *p < 0.05. Effect of *vamp33 si I. scapularis* ticks on **c** interferon (IFN)-γ, **d** interleukin (IL)-5, and **e** IL-13 in the murine skin after 3 days of feeding. Increased **f** γδ T cells, **g** IL-2 and **h–j** chemokine release at the bite site after *Vamp33 si I. scapularis* fed for 3 days on C57BL/6 mice. Mean ± SEM are plotted. Cytokine and chemokine data represent a single experiment with multiple biological replicates, whereas data with skin immune cells represent three independent experiments combined. Statistical differences in **d–j** were evaluated by two-tailed *t* test. *p < 0.05. NS not significant. *n* number of samples. Source data are provided as a Source Data file.

investigated dermal γδ T cells in the context of tick EV secretion. Dermal γδ T cells rely on the chemokine receptors CCR2 and CCR6 for cell recruitment to inflammatory sites and CCR2 for homeostatic trafficking[47]. We placed ticks injected with *vamp33* siRNA on *ccr2*$^{-/-}$ and *ccr6*$^{-/-}$ mice. Surprisingly, genetic ablation of CCR2 or CCR6 did not affect EV-dependent tick feeding (Supplementary Fig. 9), suggesting that the dermal γδ T cells were not involved in tick feeding mediated by EVs.

Subsequently, we examined the role of (Vγ5$^+$Vδ1$^+$) DETCs during tick feeding[28,29]. We observed a greater thickening of the mouse epidermis during a tick bite (Supplementary Fig. 5b), which correlated with the epidermal function of DETCs[28,29]. Therefore, we placed ticks injected with *vamp33* siRNA on wild-type mice and measured tick weight at day 3 post-attachment. We detected a significant increase in frequency of DETCs during a tick bite in the *vamp33* siRNA treatment when compared to *vamp33* scRNA control ticks (Fig. 4f). These results also agreed with our flow cytometry data showing the majority of γδ T cells in the skin bitten by the tick were TCRγδ$^{hi}$ (Fig. 4g). Conversely, the predominant population of γδ T cells in the draining lymph nodes were TCRγδ$^{lo}$, which are non-Vγ5$^+$Vδ1$^+$ T cells. Altogether, these findings strengthened our argument that DETCs are the important γδ T cell subset in the skin during a tick bite.

Then, we investigated the effect of DETCs on tick feeding by placing ticks injected with *vamp33* siRNA on FVB-Tac mice. FVB-Tac mice are naturally depleted of DETCs due to a failure of thymic selection because of a natural mutation of the *skint1* gene[52] (Fig. 4h, left-most rectangular bars). A decrease in weight was not observed when ticks deficient in EVs were placed on FVB-Tac (DETC depleted) mice (Fig. 4h, right-most rectangular bars). Conversely, we detected the opposite findings in FVB-JAX (DETC-sufficient) mice, which do not bear a mutation in the *skint1* gene and harbored normal levels of DETCs. Collectively,

we discovered that tick EVs regulate DETCs in the epidermis for an optimal feeding environment.

**Intracellular bacteria manipulate the release of tick EVs during infection.** As *I. scapularis* regulated host immunity, we examined whether intracellular bacteria altered the release of tick EVs upon infection. Accordingly, we detected a modified morphology and heterogeneity of EVs originating from tick cells previously infected with the intracellular rickettsial bacterium *A. phagocytophilum*, but not the extracellular spirochete *B. burgdorferi* (Fig. 5a and Supplementary Movies 10–12). Consistently, neither infection of tick cells with *B. burgdorferi* nor the human parasite *Babesia microti* substantially increased the secretion of EVs (Fig. 5b, Supplementary Fig. 10a, and Supplementary Movies 13 and 14). However, tick cells released more EVs in the classical exosomal range upon infection with the intracellular bacteria *Ehrlichia chaffeensis* and *F. tularensis* (Supplementary Fig. 10b, c and Supplementary Movies 15–18). Overall, we discovered that tick cells modify the production of EVs according to the intracellular bacterial lifestyle.

Oxidative stress has been recently linked to the pentose phosphate pathway (PPP) via the protein nuclear related factor 2 [NRF2][53]. Previously, we indicated that cells change their oxidation state and produce reactive oxygen species during *A. phagocytophilum* infection[54]. We observed an enrichment of proteins from the PPP and the NRF2 signaling network in EVs derived from *I. scapularis* salivary glands (Fig. 5c, Supplementary Figs. 3 and 4, and Supplementary Data 2 and 4). Therefore, we asked whether the altered cargo content of tick EVs during bacterial infection could be related to the oxidation of protein side chains (e.g., carbonylation[42]). We incubated EVs from *I. scapularis* cell lines ISE6 and IDE12 with 2,4-dinitrophenylhydrazine (DNPH), which selectively binds to carbonyl groups[55]. Carbonylation present in tick EVs were then probed with an anti-

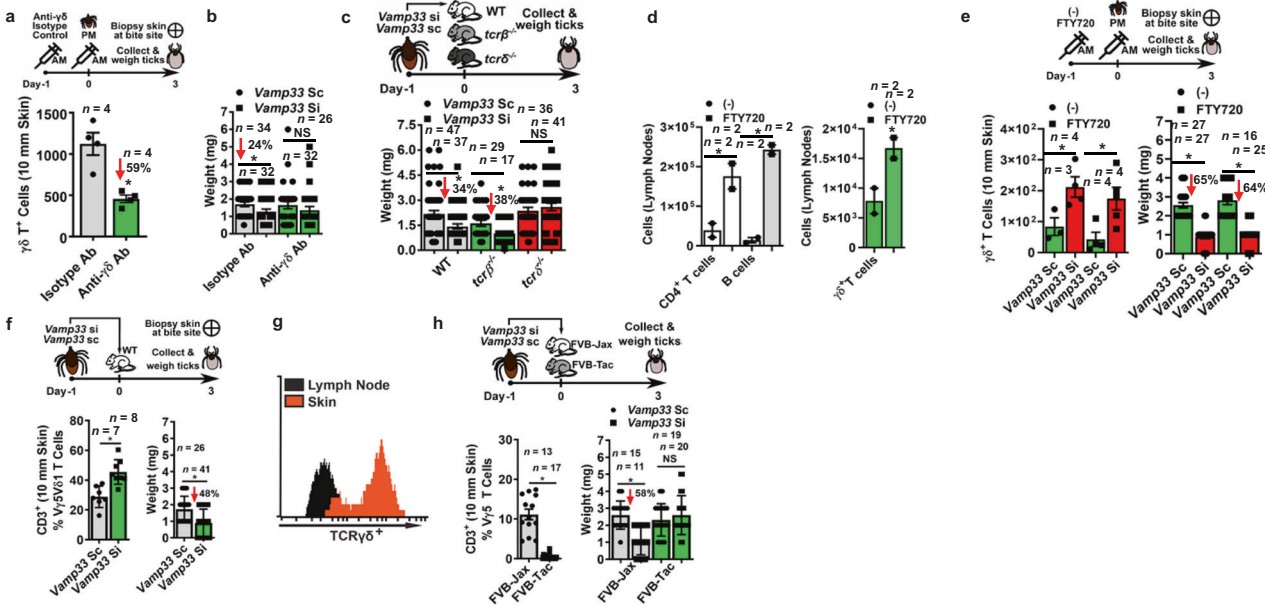

**Fig. 4 Tick EVs affect DETCs in the murine skin. a** Mice injected with anti-γδ antibodies (500 μg) display decreased γδ T cells numbers at the skin site (right). The graph is representative of two independent experiments. **b** *Vamp33* silencing (squares) resulted in similar tick bloodmeal intake when compared to the control treatment (circles) on γδ T cell-depleted mice (green). The graph represents two independent experiments combined. **c** Weight of *Vamp33* si or *Vamp33* sc ticks placed on wild type (WT; gray), *tcrβ−/−* (green), and *tcrδ−/−* (red) mice for 3 days. The graph represents four independent experiments combined. **d** Ticks were placed on FTY720-(squares) (1 mg/kg) or PBS-(circles) (−) treated C57BL/6 mice and allowed to feed for 3 days. Skin biopsies at the tick bite site and the draining lymph nodes were taken and immune cell populations were assessed by flow cytometry. The graph represents one of two independent experiments. **e** *Vamp33* sc (green) or *Vamp33* si (red) ticks were placed on PBS- (circles) (−) or FTY720- (squares)-treated (1 mg/kg) C57BL/6 mice and their weight was measured three days post-infestation (two independent experiments combined). γδ T cell graph (left) represents one of two independent experiments. **f** Ticks were placed on C57BL/6 mice and allowed to feed for 3 days and their weight was measured (two independent experiments combined). Biopsies were taken from the tick bite site and the DETC (Vγ5+Vδ1+) population was assessed by flow cytometry (one of two independent experiments). **g** Flow cytometry measurement of TCR γδhi immune population in a murine skin sample when compared to a TCR γδlo immune population in the draining lymph node. **h** Skin biopsies were obtained from FVB-Jax (gray) and FVB-Tac (green) mice. Vγ5+ T cell population was assessed by flow cytometry (two independent experiments). *Vamp33* sc (circles) or *Vamp33* si (squares) ticks were placed on FVB-Jax (gray) or FVB-Tac (green) mice. Tick weight measurement was obtained 3 days post-infestation (two independent experiments combined). Differences between treatments were evaluated with two-tailed Student's *t* test for all data sets, except one-tailed in **d**. Statistical significance for all experiments are shown *$p < 0.05$; NS not significant, *n* number of samples analyzed. Ab antibody, AM morning, PM afternoon. Source data are provided as a Source Data file. Mean ± standard error of the mean (SEM) are plotted for all experiments except in **g**.

DNPH antibody after derivatization. Interestingly, carbonylated proteins were detected in tick EVs at much higher levels in the *A. phagocytophilum*-infected treatment when compared to the control group (Fig. 5d, e). A similar effect was not observed for glycosylation (Fig. 5f) or phosphorylation (Fig. 5g). Collectively, we found out that the intracellular bacterial lifestyle modifies the cargo content and carbonylation within tick EVs.

**Tick EVs enable distinct outcomes of bacterial infection in the mammalian host**. We then asked whether tick EVs could affect microbial infection to the mammalian host. We tested this hypothesis by using two distinct models: (i) *F. tularensis*, which causes sepsis followed by death in mice[56]; and (ii) the mildly virulent rickettsial bacterium *A. phagocytophilum*, which does not cause lethality in mice[54]. We observed that *F. tularensis* injection via the intradermal route, which mimics tick infection, was fully virulent (Fig. 6a, b)[56]. Isolation of EVs from partially fed *D. andersoni* ticks, the vector of *F. tularensis* (agent of tularemia) in the western United States[38], showed a heterogeneous EV population with an average mean size of 204 ± 6 nm (Fig. 6c and Supplementary Movie 19). We then investigated whether EVs from *D. andersoni* ticks were protective against *F. tularensis*-induced sepsis. Mice injected with *D. andersoni* EVs and *F. tularensis* were more resistant to infection when compared to the control bacterial treatment (Fig. 6d, e). Mice injected with *D.*

*andersoni* EVs and *F. tularensis* had also lessened splenomegaly (Fig. 6f) and decreased levels of interferon (IFN)-γ and tumor necrosis factor (TNF)-α in the blood (Fig. 6g and Supplementary Fig. 11a). Surprisingly, bacterial numbers were reduced in the liver and the spleen, but not the lungs of mice injected with *F. tularensis* and *D. andersoni* EVs (Fig. 6h, i and Supplementary Fig. 11b).

Finally, we developed an animal model of *F. tularensis* infection in mice using the tick *D. variabilis*. The intent was to determine whether injecting $10^7$ colony-forming units (CFU) of *F. tularensis* into mice was an experimentally judicious approach. This animal model of tick infection considered the 5-day blood meal of *D. variabilis* and the rapid lethality of *F. tularensis*. Surprisingly, *D. variabilis* harbored a very large amount of *F. tularensis* with an average of $9.5 × 10^5$ CFU/tick after repletion, to a high of $6.5 × 10^6$ CFU/tick of *F. tularensis* on week 4 post-feeding prior to molting (Fig. 6j). Following these results, we then assessed *D. variabilis* infection of *F. tularensis* in the laboratory onto naïve mice. Examination of mouse blood revealed that 100% of naïve mice were infected with *F. tularensis* on day 14 after tick placement (Fig. 6k). We observed high bacterial burdens in the blood ($6 × 10^5$ CFU/mL), lungs, livers, and spleens ($3 × 10^5$ – $1.9 × 10^6$ CFU/mg tissue). Importantly, our results corroborated with a reproducible and quantitative model of *F. tularensis* in *D. variabilis*[57,58] and revealed that the presence of *F. tularensis* inside

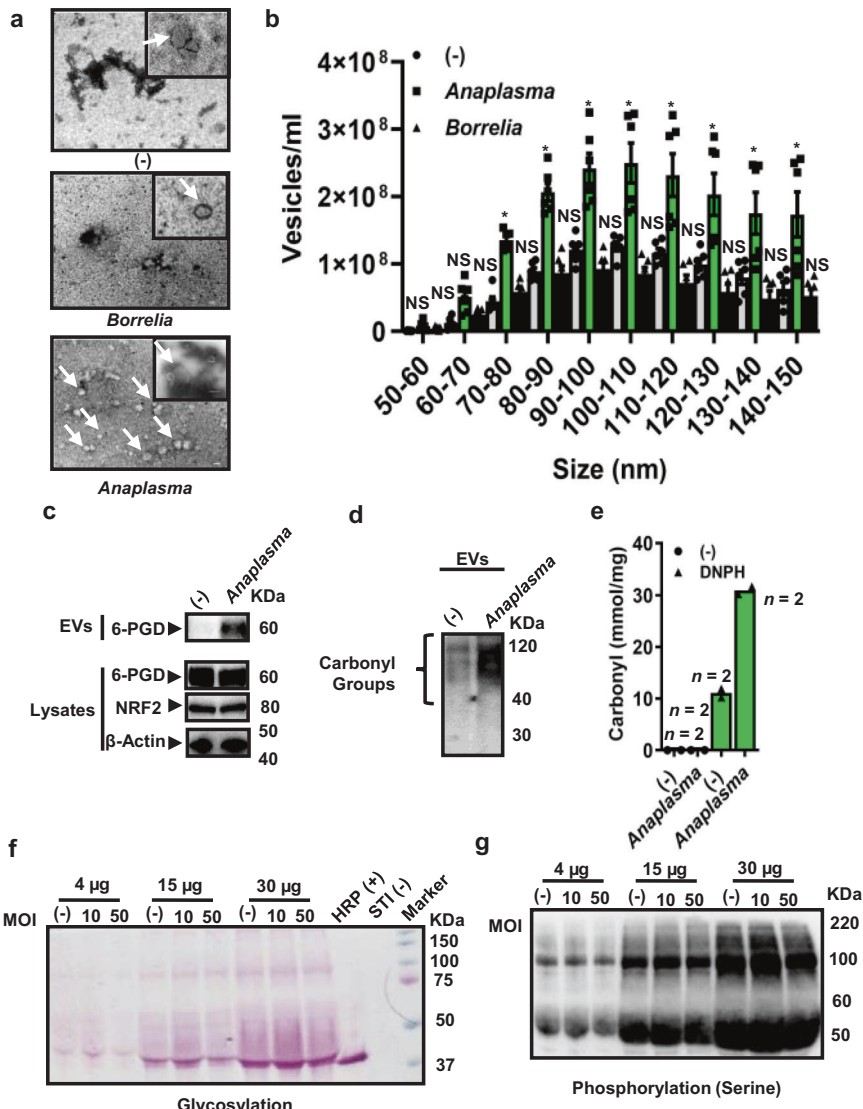

**Fig. 5 Tick EVs are modified by the intracellular bacterial lifestyle. a** Transmission electron microscopy of EVs purified from *I. scapularis* ISE6 cells infected with *A. phagocytophilum* or *B. burgdorferi*. Scale bars: 100 nm, large panel = 11,000x; small panel = 30,000x. The image is representative of two independent experiments. **b** *I. scapularis* EVs released from *A. phagocytophilum* or *B. burgdorferi*-stimulated ISE6 cells. Means ± standard error of the mean (SEM) are plotted. (−) unstimulated cells. (*$p < 0.05$). Data is representative of three technical and two biological replicates. SDS-PAGE immunoblots showing increased release of **c** 6-phosphogluconate dehydrogenase (PGD) (50 μg for lysates and 40 μl for EV immunoblots, respectively); NRF2 nuclear factor erythroid 2-related factor 2. **d** 15 μg of EVs derived from tick ISE6 cells infected with *A. phagocytophilum* (MOI 50) showing carbonylation. **e** Carbonylation in EVs derived from *A. phagocytophilum*-infected IDE12 cells (MOI 50; 15 μg of EVs for ELISA). DNPH = 2,4-dinitrophenylhydrazine. (−) represents uninfected ISE6 cells lysate. **f**, **g** glycosylation and phosphorylation of proteins in uninfected and *A. phagocytophilum*-infected (MOI 10 and 50) EVs purified from tick ISE6 cells at indicated amounts. HRP (+) represents the horseradish peroxidase positive control while STI (−) indicates the soybean trypsin inhibitor negative control. MOI multiplicity of infection. **f**, **g** blots and gels were repeated at least twice obtaining the same results. Statistics in **b** was done by using Two-way ANOVA using size and treatment as variables followed by the *post-hoc* Sidak test for comparisons. (*$p < 0.05$). Source data are provided as a Source Data file.

ticks was comparable to what has been described on Martha's Vineyard[59]. Martha's Vineyard is an island off the coast of Massachusetts where epizootic transmission of tularemia by ticks has occurred since 2001[60].

Opposite findings were observed when the mildly virulent bacterium *A. phagocytophilum* and *I. scapularis* EVs were injected intradermally into animals. *I. scapularis* EVs favored establishment of *A. phagocytophilum* to mice (Fig. 7a, b). No differences in cytokine release were observed between treatments during *A. phagocytophilum* infection (Fig. 7c, d). Collectively, our findings show that tick EVs distinctly regulate the infection of *A. phagocytophilum* and *F. tularensis* in the mammalian host during feeding.

## Discussion

In most infectious diseases, the clinical outcome of an illness is determined primarily by interactions between the pathogen and the mammalian host. Nevertheless, for vector-borne diseases, ailment is not only driven by the arms race between the pathogen and the host, but also the arthropod. Currently, arthropod EVs are thought to enable pathogen transmission to mammals[18,19]. Here, we indicate that microbial infection mediated by arthropod EVs is more intricate than previously observed. Tick EVs promoted infection of the mild rickettsial agent *A. phagocytophilum* to the mammalian host. Conversely, tick EVs decreased *F. tularensis* morbidity and mortality in mice. Importantly, tick EVs

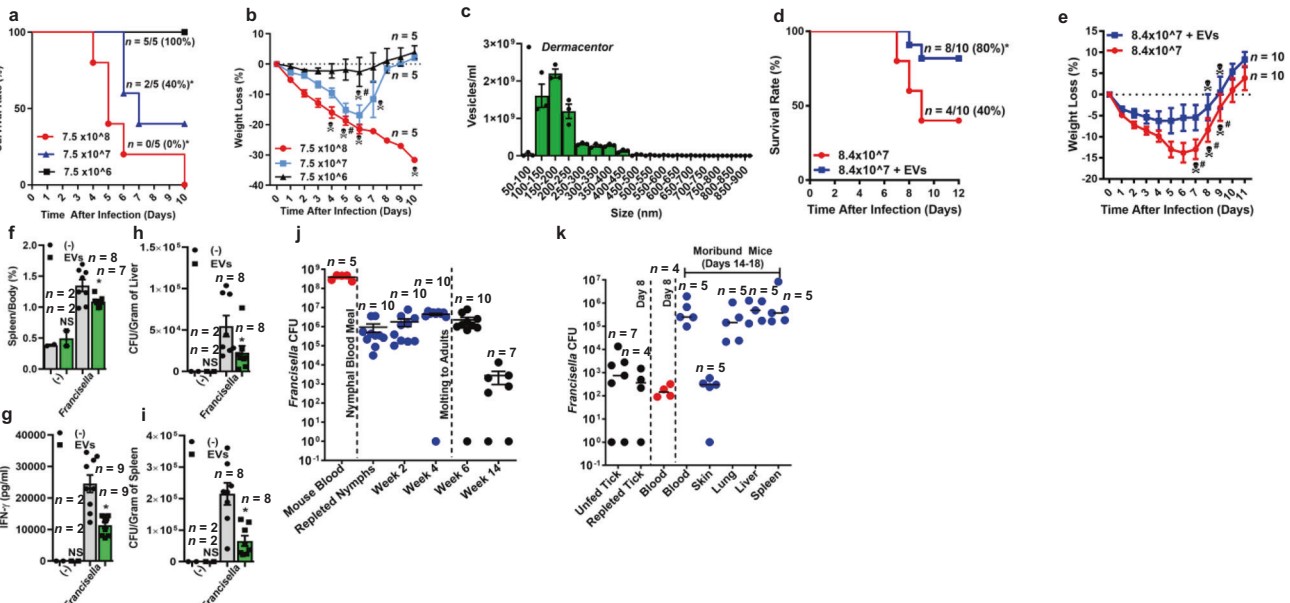

**Fig. 6 _D. andersoni_ EVs mitigate _F. tularensis_ infection in the mammalian host.** In **a** and **b** C57BL/6 mice were intradermally inoculated with three doses of _F. tularensis_ live vaccine strain: $7.5 \times 10^6$ (black), $7.5 \times 10^7$ (blue), and $7.5 \times 10^8$ (red). Survival and loss of weight were monitored daily for 10 days. Data represents the mean ± SEM; _n_ number of mice. **c** Quantity and size distribution of EVs purified from _D. andersoni_ salivary glands, as judged by nanoparticle tracking analysis. Bars represent the standard error from three technical replicates. In **d**, **e** survival and weight loss in C57BL/6 mice inoculated with $8.4 \times 10^7$ CFU of _F. tularensis_ co-injected with $1 \times 10^8$ _D. andersoni_ EVs. EV treatment (blue); control (red). For **e**, data presents the mean ± SEM. In **f–i** _F. tularensis_ ($8 \times 10^7$ CFU) was co-injected in the presence (green) or absence (gray) of _D. andersoni_ salivary gland EVs ($1 \times 10^8$). **f** splenomegaly, **g** cytokine measurement made by a multiplex cytokine ELISA and **h**, **i** plaque assays at day 5 post-infection. Data is presented as a mean and standard error of the mean (±SEM). Statistical significance was determined using a two-tailed _t_ test of _Francisella_ and _Francisella_ + EVs. **f** $p = 0.03$; **g** $p = 0.0004$; **h** $p = 0.05$; **i** $p = 0.0017$. **j** _D. variabilis_ nymphs were placed onto naïve female C3H/HeN mice on day -5 and allowed to feed for three days. Mice were intravenously infected on day $-2$ with $1 \times 10^7$ CFU of _F. tularensis_. Engorged (repleted) ticks and the mouse blood were collected 2 days later (day 0) and the CFU were obtained at indicated time points. The graph is representative of two independent experiments. **k** Infected adult _D. variabilis_ ticks (week 14) were individually placed onto naïve mice to examine _F. tularensis_ infection. Ticks completed their blood meal by day 8 and the mouse blood was harvested to quantitate bacterial numbers. Data is presented as mean. Five mice were monitored through day 18 with animals being euthanized when moribund. Skull and bones denote a single mouse death, whereas skull and bones with the pound sign indicates death of two mice. Survival was analyzed with the Kaplan–Meier curve. Statistical analysis was performed with the **a**, **d** Log-rank (Mantel-Cox) or **f–i** the two-tailed _t_ test. In **b**, **e** statistical analysis of weight data was not done due to the differential animal mortality during experiment. EVs extracellular vesicles, CFU colony-forming units. *$p \leq 0.05$. NS not significant. Source data are provided as a Source Data file.

acted locally on skin immunity. The observation that tick EVs inhibited IFN-γ and TNF-α in the mouse blood during _F. tularensis_ infection was likely a consequence of molecular cues initiated at the skin site and distally propagated to other organs[61]. Furthermore, the effect of _D. andersoni_ EVs in the spleen and the liver of mice injected with _F. tularensis_ was an outcome of disease in our animal model.

Over the past century, researchers have debated the temporal kinetics of tick transmission of _F. tularensis_ and the onset of tularemia in animal models[62–64]. Unfortunately, there is no consensus or universal approach that broadly describes the _F. tularensis_ concentration inside the tick and in the saliva during transmission. Different tick infection techniques, environmental versus laboratory experiments, animal models, tick species, developmental stages, and _Francisella_ genotype/strains lead to distinct experimental outcomes and disagreements among investigators. Thus, additional analysis will be required to determine the relevance of our findings in the natural habitat of ticks.

Correlative evidence suggested the role of SNARE proteins in arthropod feeding[65–68]. SNAREs are involved in the fusion of multivesicular bodies to the plasma membrane, releasing EVs into the extracellular milieu[15–17]. The scientific community has not yet firmly developed technical approaches to visualize and quantify EVs in vivo[15,17]. To date, only a few laboratories around the world have developed fluorescent proteins that are tagged to

the plasma membrane to observe EVs in vivo by intravital microscopy or other sophisticated imaging techniques[69,70]. Additionally, these technologies are not yet viable in ticks because there are not any genome editing tools for _I. scapularis_[39]. Finally, tick feeding and salivation occur as an intermittent process for several days on a mammalian host[71] and this uneven feeding activity changes the composition of molecules within the saliva[72,73]. Thus, the quantification of EVs or microbial numbers secreted into the bite site during feeding are not feasible at this time. Given these impediments, we determined a reasonable number of EVs to use in our assays. Experimental approaches varied widely in the literature with EVs ranging from $10^6$ to $10^8$ molecules[19,74–76]. Hence, we concluded that the administration of $1 \times 10^8$ EVs was a reasonable approach in our animal model.

The skin contains many immune cells that transport salivary antigens to the draining lymph nodes[49]. Most studies have portrayed the effects of the tick bite as taking place within the local feeding cavity and in the dermis[4,49]. However, an earlier study revealed antigens that may be trapped in the skin and affect an area beyond the dermal–epidermal location[77]. For _I. scapularis_, the effect of EVs was dependent on γδ T cells. γδ T cells act as a bridge between innate and adaptive immunity[28–30] and are important against infection by tick-borne pathogens[78,79]. γδ T cells also detect tissue damage and are involved in wound repair[28–30]. We observed that inhibition of EV biogenesis in _I._

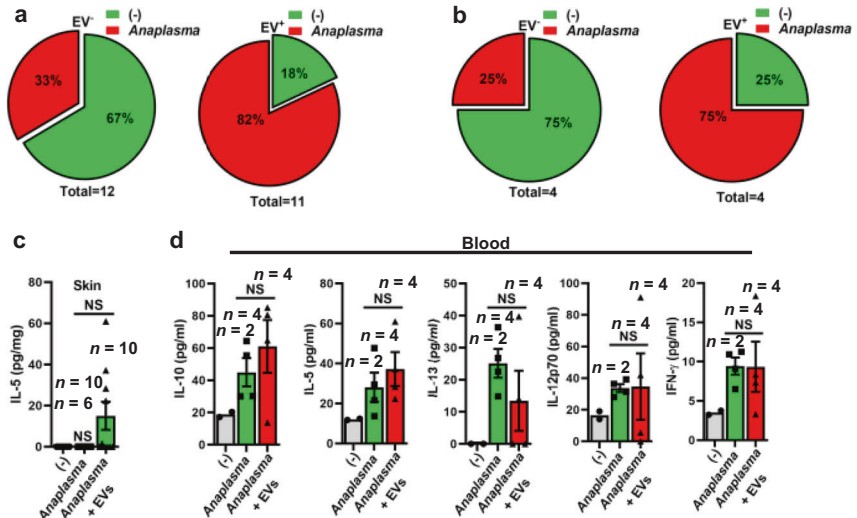

**Fig. 7 *I. scapularis* EVs enable *A. phagocytophilum* infection in the mammalian host.** *A. phagocytophilum* ($1 \times 10^7$) was intradermally injected in the presence or absence of *I. scapularis* salivary gland EVs ($4 \times 10^7$). The presence (red) or absence (green) of *A. phagocytophilum* in **a** skin or **b** spleen was measured by quantitative real-time PCR at day 3 post-infection. Cytokine measurements were done in the mouse **c** skin and **d** blood by a multiplex ELISA assay. Baseline treatment denotes skin and mouse blood without bacterial infection (gray bars). Data is presented as a mean and standard error of the mean (±SEM). (\**p* < 0.05; NS not significant). Graphs are representative of one in two independent experiments. Statistics in **c** were done by using one-way ANOVA, whereas **d** was done by two-tailed *t*-test between *Anaplasma* and *Anaplasma* + EVs. For **c** *n* = skin samples and **d** *n* = blood samples. Source data are provided as a Source Data file.

*scapularis* ticks resulted in a significant increase of γδ T cells at the bite site. Remarkably, feeding has also been shown to reduce the number of circulating γδ T cells in animals infested with the ticks *Haemaphysalis bispinosa* and *Hyalomma anatolicum*[80].

*I. scapularis* EVs affected DETCs, which are located in the mouse epidermis and important for wound healing[28–30]. DETCs interact with keratinocytes, which comprise of approximately 95% of all cells in the epidermal layer[28]. Forthcoming studies will ascertain whether tick EVs act directly on DETCs and also determine how DETCs affect cell intrinsic and extrinsic signaling in keratinocytes. For instance, tick EVs appear to change the molecular program of a human keratinocyte cell line[41]. Notably, we do not yet know whether the elongated dendrite morphology of DETCs enables contact with keratinocytes[28,81] during a tick bite. Finally, investigators need to examine the cellular composition of the epidermis in DETC-deficient (FVB-Tac) and DETC-sufficient (FVB-Jax) mice during a tick bite. FVB-Tac, but not the FVB-Jax mice, are depleted of DETCs because of an intrathymic differentiation defect due to a mutation in the *skint1* gene[52]. Under normal conditions, DETCs interact with several neighboring cells in the epidermis. Conversely, the lack of DETCs in the epidermis results in increased keratinocyte death[82] and repopulation of the epidermis by αβ T cells with a diverse T cell receptor repertoire[28].

An ecological niche is comprised of biotic and abiotic factors that dictate interspecies relationships[83]. We showed that infection with three different tick-borne intracellular bacteria, *A. phagocytophilum, F. tularensis*, and to a lesser extent *E. chaffeensis* led to an increase in EV release and changes in their cargo content. This observation has also been described for arthropod-borne viruses[18,19]; thus, this phenomenon appears to be an evolutionarily conserved mechanism. Previously, we discovered that infection with *A. phagocytophilum* increases oxidative stress[54]. Furthermore, a connection between inflammation and oxidative responses through the transcription factor Nrf2 has been established[53] and we observed that the accumulation of oxidized proteins elicits poor antigen presentation to T cells[55]. We did not detect any differences in *nrf2* expression in tick salivary glands infected with *A. phagocytophilum*. However, we observed an increased secretion of the Nrf2-regulated protein 6-PGD[53] in tick EVs. Whether the increase of 6-PGD in tick cells results from the parasitic or the mutualistic effects of *A. phagocytophilum* remains to be determined.

In summary, we used two bacterial models of infection (e.g., *A. phagocytophilum* and *F. tularensis*) to show that tick EVs promote distinct microbial outcomes in the mammalian host. Whenever possible, we recommend that arthropod vectors should be included in infectious disease studies due to their potential to modify the degree of pathogenesis in a microbe.

## Methods

**Reagents and resources.** All primers, reagents, resources, manufacturers, and catalog numbers are listed in Supplementary Data 6 and 7.

**EV-depleted medium.** L15C300 medium was supplemented with 5% fetal bovine serum (FBS; Millipore-Sigma), 5% tryptose phosphate broth (TPB; BD), 0.1% lipoprotein concentrate (LPC; MP Biomedicals), 0.25% sodium bicarbonate (NaHCO$_3$; Millipore-Sigma), and 25 mM HEPES (Millipore-Sigma)[84]. Medium was cleared from EVs by ultracentrifugation at 100,000×*g* for 18 h at 4 °C in a LE-80 ultracentrifuge (Beckman Coulter) with a 60Ti rotor. DMEM medium supplemented with 10% FBS was also freed from EVs using the same method. The absence of EVs was confirmed by determining the particle size distribution with a Zetasizer NanoZS (Malvern Panalytical) or a NanoSight NS300 (Malvern Panalytical) for nanoparticle tracking analysis (NTA). EV-free medium was sterilized by passing the content through a 0.22-μm Millipore Express® PLUS (Millipore-Sigma).

**Ticks.** *I. scapularis* ticks were obtained from three independent sources: (1) the Centers for Disease Control and Prevention through the Biodefense and Emerging Infectious Diseases (BEI) Resources Repository; (2) Dr. Utpal Pal at the University of Maryland, College Park; and (3) Dr. Ulrike Munderloh and Dr. Jonathan Oliver at the University of Minnesota. Unfed nymphs and partially engorged adult female ticks were maintained in an I-30BLL incubator (Percival Scientific) at 25 °C under saturated humidity (>85%) in 16:8 h light/dark cycle. *D. variabilis* ticks were housed in 3-dram plastic vials in glass desiccators, with 12-h light–dark cycles, at ambient temperature over saturated potassium nitrate (KNO$_3$), which generates a humidified atmosphere of >90% at 20 °C. *D. andersoni* ticks were received from the United States Department of Agriculture (USDA) – Agricultural Research Service (ARS). Partially engorged virgin females were dissected immediately upon arrival.

**Mice**. Experiments were performed on C57BL/6J (WT), *ccr2*⁻/⁻, *ccr6*⁻/⁻ *tcrdelta*⁻/⁻, *tcrbeta*⁻/⁻ FVB/N Jax, and FVB/N Tac mice. Breeding pairs were purchased from the Jackson Laboratory except FVB/N Tac mice. FVB/N Tac mice were purchased from Taconic Biosciences. C3H/HeN mice, female, 8–10 weeks old, were purchased from Charles River and used for tick-*F. tularensis* experiments. All mouse strains were bred at the University of Maryland School of Medicine, unless otherwise indicated. Male mice (6–9 weeks) were used for tick feeding, chemokine and cytokine analysis, immune cell population assays, γδ T cell antibody depletion, FTY720 treatment, and knockout experiments. Female mice (6–8 weeks) were used for bacterial infection experiments.

**Cell lines**. *I. scapularis* (ISE6), *D. andersoni* (DAE100), and *A. americanum* (AAE2) embryonic derived cell lines were obtained from Dr. Ulrike Munderloh at the University of Minnesota through a material transfer agreement. Tick cell lines were maintained in Cellstart® 25 cm² flasks (Greiner bio-one), containing L15C300 medium supplemented with 5% FBS, 5% TPB, and 0.1% LPC kept at 34 °C[84]. Bacterial infection experiments and subcultures were performed in whole 25 cm² flasks (Greiner bio-one) when cells reached confluency, as evaluated by light microscopy with an AMG EVOS Fl digital inverted microscope (Thermo Fisher Scientific). HL-60 (CCL-240) cells were obtained from ATCC and grown in RPMI-1640 medium with L-Glutamine (Quality Biological) supplemented with 10% FBS (Gemini Bio-Products) and 1% GlutaMax™ (Gibco). DH82 cells were a kind gift from Jere McBride at University of Texas Medical Branch. These cells were grown in DMEM/F12 (1:1; Gibco) supplemented with 10% FBS (Gemini Bio-Products). Both cell lines were maintained at 37 °C and 5% CO₂. Cells numbers were determined using a TC20™ automated cell counter (Bio-Rad).

**Microbial infection**. *A. phagocytophilum* strain HZ was cultured in 75 cm² flasks (CytoOne) containing HL-60 cells. Briefly, HL-60 cells were cultured in 20 ml of RPMI medium, supplemented with 10% Fetal Bovine Serum and 1x Glutamax. *A. phagocytophilum* infected cells (500 μl) were added to 5 ml of uninfected cells at 1 to $5 \times 10^6$ cells/ml diluted in 24.5 ml of media[85]. The percentage of infection was monitored by the Richard-Allan Scientific™ three-step staining (Thermo Fisher Scientific). Infected cells were spun onto microscope slides with a Cytospin 2 (Shandon). Cells were visualized by light microscopy with an Axioskop microscope (Zeiss). Bacteria were purified once cultures had reached >70% infection. Bacterial numbers were estimated using the number of infected HL-60 cells × 5 morulae/cell × 19 bacteria/cell × 0.5 (representing 50% recovery rate)[9]. Infected cells were concentrated by centrifugation at 10,000×*g* for 5 min. Cells and bacteria were resuspended in 5 ml of L15C300 EV-free medium. Bacteria were isolated from cells through a 27-G bent needle. Cell debris was separated from bacteria by centrifugation at 600x*g* for 10 mins. The supernatant was collected, and bacteria were then centrifuged at 10,000x*g* for 5 mins. Pelleted bacteria were resuspended in L15C300 EV-free media and inoculated into confluent ISE6 25 cm² flasks (Greiner bio-one) at a 50:1 MOI.

*B. burgdorferi* B31 strain, clone MSK5[86] was cultured in BSK-II supplemented with 6% normal rabbit serum (RS) until reaching mid-exponential phase. Bacteria were pelleted by centrifugation at 3000x*g* for 15 min at 4 °C and washed five times with 1x phosphate-buffered saline (PBS) to eliminate any remaining RS. Pelleted bacteria were resuspended in L15C300 EV-free media and inoculated into confluent layers of ISE6 cells at a MOI of 10:1. *E. chaffeensis* was cultured in DH82 cells[87]. DH82 cells were cultured in 75 cm² flasks until 50–80% confluency and inoculated with *E. chaffeensis* DH82 infected cells (1:10). The percentage of infection was monitored by the Richard-Allan Scientific™ three-step staining, as described for *A. phagocytophilum*. The number of bacteria was calculated using the same formula for *A. phagocytophilum*. Bacteria were purified by passing infected cells through a 27-G bent needle and using the centrifugation steps previously described for *A. phagocytophilum*. Confluent AAE2 cells were inoculated with *E. chaffeensis* (MOI 50:1) in L15C300 EV-free medium.

Frozen stocks of *F. tularensis* LVS were obtained from Dr. Katharina Richard at the University of Maryland School of Medicine. Frozen stocks were thawed and diluted in L15C300 EV-free medium. DAE100 cells were inoculated with the diluted *F. tularensis* (MOI 10) in L15C300 EV-free medium. *F. tularensis* Type B strain LVS was also obtained from BEI Resources and cultured[88]. Blood from *rag2*⁻/⁻ mice infected with *Babesia microti* LabS1 strain was collected and diluted to $10^5$ infected red blood cells (iRBCs). iRBCs were inoculated into ISE6 cells ($4 \times 10^7$ cells) flasks (Greiner bio-one) containing L15C300 EV-free medium. All tick cell line infections occurred for 18 h at 34 °C.

**Tick salivary gland culture**. Salivary gland EVs were purified from ex vivo cultures originated from partially engorged *I. scapularis* adult females and nymphs. Adult *I. scapularis* females were fed on 6–10-month-old female New Zealand white rabbits for 5–6 days at the CDC facilities and then shipped to the University of Maryland School of Medicine. For the nymph salivary gland culture, *I. scapularis* nymphs fed for 3 days on C57BL/6J (WT) mice. *D. andersoni* female adults fed on Holstein calves for a total of 6 days inside of stockinet sleeves. The stockinet sleeves were glued to the skin with a hip tag cement. Animals were stalled in moated blocks. After 24 h, ticks were placed on the skin and allowed to feed for 5 days.

Partially-fed adult female ticks (25–69) and nymphs (15–30) were dissected 1–2 days post-removal. Ticks were dissected with 4 mm vannas scissors (Fine Science Tools) under a Stereo Zoom stereoscope (Bausch and Comb). In all, 10 μl of PBS were added to samples to avoid desiccation. Midguts, Malpighian tubes, and other organs were removed. Salivary glands were dissected and cultured in 24-well culture plates (Corning). 10 salivary glands from adult ticks or 20 salivary glands from nymphs were placed in each well, containing 500 μl of L15C300 EV-free medium supplemented with 1x penicillin/streptomycin (Corning) and 1x Amphotericin B (Gibco). Salivary glands were incubated for 24 h at 34 °C to allow EV secretion.

To study the effect of intracellular bacteria on the content of salivary EVs, salivary glands (70–80) from adult female ticks (35–40) were dissected, as described above. Ten salivary glands were placed per well in vesicle-free medium supplemented with Amphotericin B (Gibco). Cell-free *A. phagocytophilum* was isolated and inoculated to the salivary cultures ($1.4 \times 10^8$ bacteria/well). Salivary glands were incubated for 24 h at 34 °C in the presence or absence of bacteria.

**Tick saliva**. Tick saliva was collected as described by Hackenberg et al.[89]. Briefly, fully fed *Ixodes ricinus* females (6–7 days) were fixed onto glass slides. Salivation was induced by applying 2 μl of pilocarpine solution (33 g/l in 96% ethanol; Sigma-Aldrich) to the tick dorsal scutum. Saliva was collected into capillary tubes placed around the hypostome of each tick. Ticks were maintained within a humidified chamber in sterile conditions at 37 °C for 2–3 h during salivation. The saliva was pooled, filtered through a 0.22-μm filter, and stored in −80 °C before being shipped to the University of Maryland School of Medicine.

**EV purification**. Medium collected from cell or salivary gland cultures were cleared of any live cells by centrifugation at 300 × *g* for 10 min at 4 °C. Dead cells were removed by a second centrifugation at 2000 × *g* for 10 min at 4 °C. The supernatant was collected, and apoptotic bodies were removed by a third centrifugation at 10,000 × *g* for 30 min at 10 °C. To reduce the number of EVs >200 nm in size, the supernatant was filtered through a 0.22-μm Millipore syringe filter (Millipore-Sigma). EVs were pelleted by ultracentrifugation (100,000 × *g*) for 18 h at 4 °C. Supernatant was discarded and EVs were resuspended in PBS. Salivary EVs were purified using a protocol modified from Michael et al.[90] and Zlotogorski-Hurvitz et al.[91]. In brief, tick saliva was centrifuged at 1500 × *g* for 10 min at 4 °C to discard any cells present in the sample. The supernatant was collected and centrifuged for 20 min at 12,000 × *g* at 4 °C. Following this procedure, the recovered material was diluted 1:1 in 1x PBS to reduce viscosity and centrifuged at 17,000 × *g* for 15 min at 4 °C. The supernatant was collected and the EVs were purified by ultracentrifugation at 100,000 × *g* for 18 h at 4 °C. EVs were resuspended in RIPA buffer for western blotting.

**Transmission electron microscopy**. ISE6 cells were infected with either *A. phagocytophilum* or *B. burgdorferi*. EVs from equal number of infected or uninfected cells ($4.44 \times 10^7$) were purified as described above and resuspended in PBS. EVs were further concentrated with Amicon® Ultra 0.5 ml centrifugal filters 30 K (Millipore-Sigma). Samples were glow-discharged with 400 mesh formva-coated copper grids (Electron Microscopy Sciences) and negatively stained with 1% uranyl acetate (UA) or 1% sodium potassium phosphotungstate (PTA). Air-dried grids were examined in a Tecnai T12 transmission electron microscope (Thermo Scientific) at an operating voltage of 80 keV. Images were acquired using a bottom mount CCD camera (Advanced Microscopy Techniques) and AMT600 software (Advanced Microscopy Techniques). Transmission electron microscopy images were analyzed using ImageJ.

**EV quantification**. EV concentration and sizes were determined using the NanoSight LM10 or NS300 (Malvern Panalytical) machines with NTA software versions 2.0 or 3.0, respectively. The mean of the size generated in the NTA reports was used to calculate the average size of the EVs in each sample. Data was analyzed using GraphPad Version 9.1.0 from Prism.

**Proteomics of EVs**. EVs from adult *I. scapularis* female salivary gland cultures were concentrated in L15C300 EV-free medium using a 300 kDa Vivaspin500 filter (Sartorius). Concentrated EVs were re-suspended in 0.95 M sucrose (Sigma S0389) and applied on top of the following sucrose gradient: 2 M, 1.65 M, 1.3 M, 0.95 M, 0.6 M, 0.25 M in a polypropylene ultracentrifuge tube. Samples were ultracentrifuged at 151,000x*g* for 16 h at 4 °C. The first 750 μl of the sample were removed from the top of the gradient as waste. The next two 1.5 ml fractions were collected, washed with PBS, and ultracentrifuged for 70 min at 77,100 × *g* at 4 °C. Supernatant was removed and the pellet was processed.

*1D SDS-PAGE and in-gel trypsin digestion*. EV pellets were processed for proteomics analysis after fractionation on a 4–12% gradient NuPAGE™ Bis-Tris gel (Invitrogen). Gel was stained with Silver Stain for Mass Spectrometry Kit (Pierce) and pieces of the gel were cut out, de-stained and washed. Gel pieces were reduced at 55 °C for 1 h with 20 mM DTT in 25 mM ammonium bicarbonate buffer and alkylated for 50 min at room temperature in the dark with 55 mM iodoacetamide

in 25 mM ammonium bicarbonate buffer. Proteins were digested with trypsin and Lys-C mix (Promega) at 37 °C overnight. Following digestion, samples were evaporated to 100 μl and desalted and concentrated with a C18 ZipTip (Millipore).

*NanoLC–ESI–MS/MS analysis of tryptic/LysC peptides.* Peptides were sequenced and analyzed by nanoLC–ESI–MS/MS using Qexactive HF quadrupole orbitrap mass spectrometers (Thermo Fisher Scientific), which was coupled to an Easy nLC 1000 UHPLC (Thermo Fisher Scientific) through a nanoelectrospray ion source. Peptides were separated on a C18 analytical column (100 μm internal diameter, × 20 cm length) packed with 2.7 μm Phenomenex Cortecs particles. The solution of peptides was equilibrated with 3 μl 5% acetonitrile 0.1% formic acid, before being separated using optimized 180 min linear gradients employing 2–32% acetonitrile in a buffer with 0.1% formic acid, at a flow rate of 400 ηl/min. The LC mobile phase solvents and the sample dilutions contained 0.1% formic acid in water (Buffer A) and 0.1% formic acid in 100% acetonitrile (Buffer B) (Optima™ LC/MS; Fisher Scientific). Data acquisition was performed using the instrument provided Xcalibur™ (version 3.0) software. The mass spectrometer was operated in the positive ion ionization and data–dependent acquisition (DDA) mode. The full MS scans were obtained with a range of m/z 300 to 2000, at a mass resolution of 120,000 at m/z 200, and a target value of $1.00E + 06$ with the maximum injection time of 50 ms. HCD collision was performed on the 15 most significant peaks, and tandem mass spectra were acquired at a mass resolution of 15,000 at m/z 200 and a target value of $1.00E + 05$ with the maximum injection time of 110 ms. The dynamic exclusion time was 20 s. The normalized collision energy was optimized for 32–34%.

*Protein identification.* Raw files were "de novo" sequenced and assigned with a protein ID using the Peaks 8.5/X+software (Bioinformatics Solutions, Waterloo, Canada) by searching against the "I. scapularis" genome in SwissProt database (January 2018; 21974 entries and updated December 2020 TrEMBL 62856 entries (38 reviewed by SwissProt). The following search parameters were applied for protein ID analysis: trypsin, Lys-C, as restriction enzymes and two allowed missed cleavages at both peptides ends. For the data acquired on QEHF, the parent mass tolerance was set to 15 ppm using monoisotopic mass and the fragment ion mass tolerance was set to 0.05 Da. Carbamidomethyl cysteine (+57.0215 on C) was specified in PEAKS 8.5/X + as a fixed modification. Methionine, histidine, and tryptophan oxidations (+15.99 on MHW), deamidation of asparagine and glutamine (NQ-0.98), and pyro-Glu from glutamine (Q-18.01 N-term) were set as variable modifications.

To account for the possible redox stress induced post-translational modifications, we applied additional variable modification: Lys->Allysine (−1.03; K); Lys->Amino adipicAcid (+14.96; K); Amino (Y) (+15.01, Y); Trp->Oxolactone (W) (+13.98, W); Pro->Pyrrolidone (P) (−27.99; P); Pro->Pyrrolidinone (P) (−30.01; P); Arg->GluSA (−43.05; R); Dehydrated 4-hydroxynonenal (HNE-Delta:H(2)O(H)) (+138.21, C,H, K); reduced 4-Hydroxynonenal (HNE + Delta:H(2) (H)) (+158.13, C,H,K); HNE (+156.12, C,H,K); 4-Oxononenal (ONE) (4-ONE) (+154.10, C,H,K); dehydrated 4-Oxononenal Michael adduct (4-ONE + Delta:H(−2)O(−1)) (+136.09; C,H,K); Carboxymethylation (CML, CMA) (+58.005; K, R); and Carboxyethylation (CEL) (+72.021; K, R). Additional variable post-translational modifications were input during the searches to map possible changes in the signal transduction pathways involving phosphorylation (+79.97 on STYD), sulfation (+79.96 on TYS), ubiquiti nation(+114.04 on KST), and methylation (+14.02 on DEHST).

All MS data were first validated using the false discovery rate (FDR) method built in PEAKS 8.5/X + (which uses the decoy fusion method). Protein identifications were accepted if they could be assigned a confidence score (−10lgP) >20 and (−10lgP)>20 for peptides (corresponding to p-values < 0.01). The false discovery rate for proteins was further adjusted to <3% after removing contaminants like keratins and allowing a minimum of 1 peptide per identified protein. An independent validation of the MS/MS-based peptides and protein identification was performed with the Scaffold (version Scaffold_4.6.2 and 4.10.0, Proteome Software Inc.) using the compatible ".mzid" files of each sample exported from PEAKS versions 8.5 and X+. Peptide identifications were accepted if they could be established at >90.0% probability by the Scaffold Local FDR algorithm. Protein identifications were accepted if they could be established at >90.0% probability and contained at least 1 identified peptide. Protein probabilities were assigned by the Protein Prophet algorithm[92]. Proteins that contained similar peptides and could not be differentiated based on MS/MS analysis alone were grouped to satisfy the principles of parsimony. These settings ensured the attained 0.13% Decoy FDR for peptides and 0.7% FDR for proteins.

All proteins identified by 1 unique MS/MS and 1 unique peptide were further validated manually using the following criteria: (1) series of at least at least 3 or 4 b and y ions, altogether covering more than 80% of the identified peptide sequence; (2) correct assessment of fragments due to the neutral loses of $H_2O$ (18 Da), carbonyl (CO) (28 Da) (i.e., "a" ions for "b" series) and loss of phosphates (98 Da for H3PO4 and 80 Da for HPO3); (3) correct assessment of internal fragments, mainly due to the presence of proline in the sequence. Proline residues normally fragment N-terminal to the residue, producing fragments in either direction; (4) correct assignments for the m/z and charge state of both the precursor and main a 'b' and "y" ions by inspecting the corresponding isotope envelopes. The MS/MS views together with the identified a, b and y ions were exported from PEAKS and

are shown in the Supplementary Data 3 for all protein hits identified with at least 1 unique peptide.

*Functional analysis.* Proteins were annotated with GO terms from NCBI (downloaded on March 2016). Networks, functional analyses, and biochemical and cellular pathways were generated by employing the ingenuity pathway analysis (IPA; Ingenuity Systems) (Supplementary Data 5). For all IPA analysis, we used the identified proteins with their corresponding IDs presented in Supplementary Data 2. These molecules were overlaid onto a global molecular network provided by the Ingenuity Knowledge Base. The gene-ontology (GO) enrichment, networks and canonical pathways were then algorithmically generated based on their connectivity index using the built-in IPA algorithm. The probability of having a relationship between each IPA indexed biological function and the experimentally determined genes was calculated by a right-tailed Fisher's exact test with Benjamini–Hochberg multiple-testing correction. The level of significance was set to p < 0.05. Accordingly, the IPA analysis identified the molecular and cellular pathways from the IPA library of canonical pathways that were most significant to the dataset (−log (p value) > 1.5).

*EV populations.* Proteins identified in I. scapularis salivary EVs were blasted against human homologs using PSI-BLAST (https://blast.ncbi.nlm.nih.gov). The gene code and protein ID were obtained for each tick protein, and their respective human homolog from Uniprot (https://www.uniprot.org/). These gene codes were used to determine to the enrichment of proteins from exosome and/or microvesicle origin by comparing the list of gene codes obtained from tick salivary vesicles with proteins reported in Vesiclepedia (http://microvesicles.org/#) using FunRich (Functional and Enrichment analysis tool; http://funrich.org/index.html). Proteins that did not have human homologs or that their gene code was not recognized were not used for this analysis. Proteins that shared the same human homolog were included only once (Supplementary Data 4). The presence of markers for specific EV populations (Exo-S, Exo-L, and Exomeres) was determined by comparing the I. scapularis salivary EVs data set with the markers proposed elsewhere[40].

**Western blot analysis**. Protein concentrations in EVs and cell lysates were determined with the Pierce™ BCA Protein Assay kit (Thermo Fisher Scientific). To assess the enrichment of exosomal markers in tick EVs, vesicles were purified from uninfected or infected tick cell lines, salivary glands, and I. ricinus saliva, as described above. Proteins were electrophoresed in 4–15% Mini-PROTEAN® TGX™ precast gels (Bio-Rad) for 1.5 h at 100 V and transferred to 0.2 μm PVDF membranes (Trans-blot® Turbo™ transfer pack; Bio-Rad) for 30 min in a TransBlot® Turbo™ transfer system (Bio-Rad). Membranes were blotted in 5% dry-milk in PBS for 1 h at RT. Membranes were incubated with anti-TSG101 (1:500 dilution; System Biosciences EXOAB-TSG101-1), -CD63 (1:500 dilution; System Biosciences EXOAB-CD63A-1), -ALIX (1:500 dilution; System Biosciences EXOAB-ALIX-1), -SL2 (1:500 dilution), -Calnexin (1:10,000 dilution; Millipore-Sigma AB2301), -CCT7 (1 ng/μl; Proteintech 15994-1-AP), -CTNNB1 (1 ng/μl; Invitrogen MA1300), -DHX16 (1 ng/μl; Invitrogen PA530272), -PLS3 (1 ng/μl; Sigma SAB2700266-100UL), -GFPT1 (1 ng/μl; Proteintech 14132-1-AP), or -phospho-serine (1 ng/μl; Abcam ab9332-100 μg) antibodies overnight at 4 °C. Bands were detected by incubating the membrane with anti-rabbit (Novex A16023) or mouse-HRP-conjugated antibodies (Abcam ab99632; 1:4000) for 1 h at RT, labeled with Pierce® ECL Western Blot Substrate (Thermo Fisher Scientific) and visualized with a ChemiDoc™ MP (Bio-Rad). Images of the blot were acquired with Image Lab Software version 5.0 (Bio-Rad). Equal numbers of ISE6 cells ($3.46 \times 10^7$ cells) were infected with either A. phagocytophilum or B. burgdoferi. Proteins were run in western blots.

**EV glycosylation assays**. EVs were purified from partially engorged I. scapularis salivary glands and uninfected or A. phagocytophilum (MOI 10 and 50)-infected tick ISE6 and IDE12 cells. Protein concentrations in EVs were determined with the Pierce™ BCA Protein Assay kit (Thermo Fisher Scientific). Proteins were loaded at an equal amount and electrophoresed in 4–15% Mini-PROTEAN® TGX™ precast gels (Bio-Rad) for 1.5 h at 100 V followed by glycosylation assay with Pierce Glycoprotein Staining Kit (Thermo Fisher Scientific).

**EV oxidation assays**
*Western blot.* Carbonylated proteins within tick EVs were detected using the OxiSelect™ Protein Carbonyl Immunoblot kit (Cell Biolabs STA-308-T). EVs from partially engorged I. scapularis salivary glands and uninfected or Anaplasma (MOI 10 and 50)-infected tick ISE6, and IDE12 cells were purified and resuspended in PBS. Protein concentrations were measured with the Pierce BCA Protein Assay Kit. Equal amounts of protein (15 or 30 μg) were loaded onto 4–15% Mini-PROTEAN® TGX™ precast gels, electrophoresed for 1.5 h at 100 V, and transferred into 0.2 μm PVDF membranes (Trans-blot® Turbo™ transfer pack) for 30 min at 25V-1.0 A in a TransBlot® Turbo™ transfer system. Membranes were submerged into 100% methanol, dried for 5 min at RT, equilibrated in Tris Buffered Saline (TBS) with 20% methanol (Millipore-Sigma) for 5 min, and washed in 2 N HCl for 5 min, as recommended by the manufacturer. The membrane was then incubated in 1x DNPH in 2 N HCl for 5 min at RT and washed three times in 2 N HCl for 5 min.

The DNPH and non-DNPH (control)-treated membranes were blotted in 5% dry-milk in PBS-T for 1 h at RT. Carbonylated proteins were detected with rabbit anti-DNP antibodies (1:1000) for 2 h at RT and with anti-Rabbit-HRP-conjugated antibodies (1:1000) for 2 h at RT.

*ELISA*. We purified EVs from $4.10 \times 10^7$ uninfected and *A. phagocytophilum*-infected ISE6 cells. Protein carbonylation was confirmed using the OxiSelect™ Protein Carbonyl ELISA kit (Cell Biolabs STA-310). Protein concentrations were measured with the Pierce BCA Protein Assay Kit. In all, 1 µg of protein per well was loaded into a 96-well plate and allowed to bind overnight at 4 °C. Wells were washed three times with 1x PBS and 1x DNPH working solution, incubating the wells for 45 min at RT. Wells were washed five times with 1:1 ethanol:PBS and two times with 1x PBS. Wells were then blocked for 2 h at RT and washed, following the manufacturer's recommendations. Anti-DNP antibodies were added to each well and the plate was incubated for 1 h at RT under constant shaking. After three washes, HRP-conjugated antibodies were added to each well and incubated for 1 h at RT. Wells were washed five times and protein carbonylation was detected with 1x substrate solution. The reaction was blocked by adding 100 µl of the stop solution and absorbance was read in an iMark microplate reader at 450 nm (Bio-Rad).

*PGD and NRF2 western blots*. Vesicle blots were probed with anti-NRF2 (GeneTex GTX55732; 1:1000) and anti-PGD (Thermo Fisher Scientific PA5-27486; 1:1000) polyclonal antibodies. Bands were stained as described above.

## Macrophage differentiation

*Murine*. Bone marrow-derived macrophages (BMDMs) were generated from C57BL/6J mice[40]. In brief, mice were euthanized using $CO_2$ and femurs were dissected. Bone marrow was flushed by injecting differentiating medium consisting of DMEM supplemented with 30% L929 condition medium, 10% FBS (Gemini Bio-products), 1x penicillin/streptomycin (Corning), and 1x Amphotericin B (Gibco) into one end of the femur with a 25G needle. Cells were seeded onto 90 mm Petri dish plates and incubated at 37 °C 5% $CO_2$. Additional differentiating medium was added to cells on day 3 after seeding. Cells were incubated for 7 days until completely differentiated.

*Human*. Human macrophages were differentiated from peripheral blood mono-nuclear cells (PBMCs) as described in Noel et al.[93]. Briefly, human blood was obtained from healthy adult volunteers. PBMCs were purified from 10 ml of EDTA-treated human blood using Ficoll-Paque PREMIUM density 1.007 g/ml (GE Healthcare). Contaminating red blood cells were lysed with ACK lysis buffer for 5 min at room temperature (RT). Monocytes were enriched by negative selection using the human Pan Monocyte Isolation Kit (Miltenyibiotec) and LS Columns (Miltenyibiotec). Monocytes were resuspended in RPMI supplemented with 10% FBS (Gemini Bio-products), 55 µM 2-Mercaptoethanol (Gibco), 1 mM Sodium Pyruvate (Gibco), 1x MEM non-essential amino acids (Gibco), and 1x penicillin/streptomycin (Corning). Cells were counted by Trypan blue stain (0.4%; Thermo Fisher Scientific) in a TC20™ Automated cell counter (Bio-Rad) and $2 \times 10^6$ monocytes were seeded into six-well plates (Sigma). Human recombinant Macrophage Colony-Stimulating Factor (M-CSF; Biolegend) was added to each well at 50 ng/ml final concentration on days 0, 2, and 4 after seeding. Medium was changed on day 4. Cells were incubated at 37 °C 5% $CO_2$ for 6 days to allow differentiation into M0 macrophages.

## EV labeling

*PKH26 staining*. Tick cell EVs were stained as described in van der Vlist et al.[94]. Staining solution was prepared by mixing 1.5 µl of PKH26 with 100 µl diluent C in a microcentrifuge tube. PKH26 is a red fluorescent dye with long aliphatic tails[45]. In Fig. 2c, g, PKH26 was artificially transformed to an orange color to be visualized by color-blind readers. EVs (5 µg) were resuspended in 20 µl of 0.2% BSA in PBS precleared of protein aggregates by overnight ultracentrifugation at $100,000 \times g$ at 4 °C and diluted into 80 µl diluent C. EVs were transferred to the staining mix, mixed by pipetting, and incubated at RT for 3 min. The labeling was stopped by adding 100 µl phenol red-free IMDM. Protein aggregates and extra dye was separated from the labeled EVs with a 300-kDa Vivaspin500 filter (Sartorius). EVs were washed three times with IMDM and resuspended in phenol red-free IMDM.

*DiO staining*. Tick cell EVs were labeled by adding 2.5–5 µl of Vibrant DiO cell-labelling solution (Invitrogen) to 480 µg of EVs in 500 µl of EV-free DMEM medium and incubated for 20 min at 37 °C. Excess label and protein aggregates were cleared through a 300-kDa Vivaspin500 filter (Sartorius). EVs were washed three times with EV-free DMEM and resuspended in 150 µl of EV DMEM medium.

## EV-macrophage interactions

*Confocal microscopy*. BMDM and human macrophages were isolated as described above. Cells ($3 \times 10^5$) were seeded onto 35 mm glass bottom dishes (MatTek Corporation) and allowed to attach overnight. Medium was replaced using phenol red-free, EVs-free DMEM medium and 5 µM Cytochalasin D (Santa Cruz

Biotechnology). Cells were incubated for 30 min at 37 °C under 5% $CO_2$. PKH26-labeled EVs (25 µg/MatTek) were added after incubation with Cytochalasin D and cells were incubated for 4–5 h at 37 °C under 5% $CO_2$ to allow binding. Control cells did not receive EVs. Cell plasma membrane was labeled with CellMask™ Green plasma membrane stain (1:1000; Invitrogen) at 37 °C for 5 min. The medium was replaced with phenol red-free, EV-free DMEM medium containing 5 µM Cyto-chalasin D. Z-stacked images were acquired with a LSM 5 Live DUOScan laser scanning microscope (Zeiss). Composites were constructed using ImageJ (NIH) Z projection (maximum intensity).

*Live-cell confocal microscopy*. BMDMs were seeded and treated as described above. Cells were labeled with CellMask™ Green plasma membrane stain (Invitrogen). Medium was replaced with phenol red-free, EV-free DMEM medium containing 5 µM Cytochalasin D. Cells were placed in an external unit and kept at 37 °C under 5% $CO_2$. Z-stacked images were acquired every 5 min for 2 h with a LSM 5 Live DUOScan laser scanning microscope (Zeiss). Twenty-five µg of PKH26-labeled EVs were added after time 0 for the positive control. No EVs were added to the negative controls. Z-projections (maximum intensity) of each channel and videos of the Z projections were constructed in ImageJ at 2 frames-per-second (fps). Arrows were added with Windows Video Maker (Windows).

*Time-lapse live-cell microscopy*. BMDMs were seeded and labeled with CellMask™ green plasma membrane stain (Invitrogen). After 5-min staining, the medium was replaced with phenol red-free, EVs-free DMEM medium, and the MatTek flask was placed into a Viva View FL incubator (Olympus) at 37 °C under 5% $CO_2$. PKH26-labeled EVs (40 µg) were added before collecting images. Images were acquired with an Orca $R^2$ digital CCD camera (Hamamatsu) and the MetaMorph software every 5 min for 7 h. Images were grouped into stacks using ImageJ for each fluorescence channel. The change in the arbitrary fluorescence units (AFU) was measured with ImageJ, as follows: cells (7 positions) were chosen using the "freehand selection" in the green channel. Cell selection was moved to the red channel using the "ROI manager". The increase in red fluorescence was measured with the "Plot Z-axis Profile" option. The same procedure was repeated with 7 positions close to the cells to measure the fluorescence in the background. The "Raw Intensity" of the background was subtracted from the "Raw Intensity" of the cell for each time point. The resulting "Raw Intensity" of each time point was normalized by the "Raw Intensity" at time 0 to account for differences in back-ground between the control and other samples.

*Flow cytometry*. BMDMs and human macrophages were differentiated, as described above. Differentiated macrophages were resuspended in EV-free DMEM medium and aliquoted into microfuge tubes. Cells ($5 \times 10^5$) were incubated with 40 µg of DiO-labeled EVs, 40 µg of unlabeled EVs, or a combination of 20 µg of DiO-labeled and 20 µg of unlabeled EVs, or no EVs for 45 min at 37 °C under 5% $CO_2$. Cells were centrifuged at $300 \times g$ for 10 min to eliminate unbound EVs. Macrophages were resuspended in 250 µl of EV-free DMEM medium and labeled with either APC-labeled anti-mouse F4/80 antibodies (Invitrogen MF48005; 1:100) or APC/Cy7-labeled anti-human CD11b (Biolegend 301341; 1:100) and APC anti-human CD14 (eBioscience 17-0149-41; 1:100) for 30 min at RT. Three replicates were left unlabeled as controls. Excess antibody was eliminated by centrifuging at $300 \times g$ for 10 min and resuspending cells in 1% paraformaldehyde in PBS. Cell fluorescent labeling was analyzed using a LSR II (BD Biosciences). Data was analyzed with FCS Express 6 Flow Research Edition (De Novo Software). Experiments were performed in the Flow and Mass Cytometry Core Facility of the University of Maryland School of Medicine Center for Innovative Biomedical Resources (CIBR).

## RNAi silencing

Tick homologs of proteins involved in exosome biogenesis in human cells (Supplementary Data 1) were determined using PSI-BLAST. These homologs were used to re-construct the exosome biogenesis pathway in the *I. scapularis* genome. Small interfering RNAs (siRNAs) and scramble RNAs (scRNAs) were designed for two v-SNARES, *synaptobrevin 2* (*vamp2*) and *vamp33* (NCBI reference in Supplementary Data 6). Silencing (Si)RNAs were designed using BLOCK-iT™ RNAi designer (http://rnaidesigner.thermofisher.com/rnaiexpress/). Scramble (Sc) RNA was designed using GenScript (https://www.genscript.com/tools/create-scrambled-sequence). Both Si and ScRNAs were blasted against the *I. scapularis* genome to avoid off-target effects and synthesized according to the Silencer® SiRNA construction kit (Thermo Fisher Scientific). Primers are described in Supplementary Data 6.

*In vitro silencing*. ISE6 (~$2 \times 10^7$) cells were placed in 25 cm² flasks and transfected with 30 µg of SiRNA or ScRNA against *synaptobrevin 2* or *vamp33* diluted in 500 µl of normal L15C300 medium, and 25 µl Lipofectamine 2000 (Thermo Fisher Scientific) diluted in 475 µl normal L15C300 medium (1:40). Cells were incubated with the transfection agent for 24 h at 34 °C and medium was replaced by EV-free medium. Cells released EVs for 18 h at 34 °C. EVs were purified as described above. The viability of cells was measured using Trypan blue stain. EV concentration was measured using NTA. After silencing, cells were pelleted and resuspend in 2 ml TRIzol® reagent (Thermo Fisher Scientific) and stored at −80 °C until RNA isolation. RNA was isolated from 600 µl of the TRIzol suspension using PureLink™

RNA mini kit (Thermo Fisher Scientific). RNA was measured using a NanoDrop Spectrophotometer ND-1000 (Thermo Fisher Scientific). cDNA was synthesized from 150 to 400 ŋg (5 µl) of RNA using Verso cDNA synthesis kit (Thermo scientific).

*In vivo silencing*. Unfed nymphs were microinjected with 50-80 ŋg of Si or ScRNA against *synaptobrevin 2* or *vamp33* using a Nanoject III (Drummond Scientific Company). Ticks recovered overnight at 25 ℃ with saturated humidity. Live ticks were placed on mice and allowed to feed for 3 days. Ticks were stored at −80 ℃ for RNA purification. To purify mRNA, ticks were fast-frozen in liquid nitrogen and crushed with small plastic pestles. TRIzol® reagent (200 µl) was added to the crushed tick and RNA was purified using the PureLink™ RNA mini kit. cDNA was synthesized from 50 to 200 ŋg (5–10 µl) of RNA using Verso cDNA synthesis kit (Thermo scientific).

**Gene expression**. Primers to determine the expression levels of *synaptobrevin 2*, *vamp33*, and *I. scapularis actin* (Supplementary Data 6) were designed using Geneious R10 (Biomatters Inc.) and checked for specificity with Primer BLAST (https://www.ncbi.nlm.nih.gov/tools/primer-blast/). Standard curves were produced using the pCR®4 plasmid. To determine copy numbers, the mass of the plasmid with each amplicon was determined using the Sequence Manipulation suite DNA molecular weight (http://www.bioinformatics.org/sms2/dna_mw.html) and the copy number calculator (http://scienceprimer.com/copy-number-calculator-for-realtime-pcr). Levels of gene expression were measured by absolute quantification using Power SYBR® Green PCR master mix (Thermo Fisher Scientific) and normalized to the expression of tick *actin*. Amplifications were done using the following conditions: a denaturation cycle of 10 mins at 95 ℃, followed by 40 cycles of denaturalization for 15 s at 95 ℃, amplification at two different annealing temperatures (*synaptobrevin* and *vamp33*: 51 ℃; tick *actin*: 57 ℃) for 1 min. Primers were used at a final concentration of 400 ŋM each and 2 µl of cDNA were used as template. The specificity of the products was determined by single peaks in the melting curves. Quantitate PCR was collected in the Applied Biosystems 7500 Fast Real-Time PCR System through the 7500 Software v2.3.

**Weight measurements**. Ticks were microinjected with silencing RNA (*vamp33 si* and *synap2 si*) or scramble RNA (*vamp33 sc* and *synap2 sc*). Surviving ticks were placed on C57BL/6 J (WT), FVB/N Jax, FVB/N Tac, or knockout mice to feed for 3 days. Ticks were recovered from a water trap (fully engorged) or were removed with forceps (partially engorged). The percentage of attachment was calculated from the total number of ticks recovered divided by the total number of ticks applied. The weight of the ticks was measured using a Pioneer™ analytical balance (OHAUS) and the differences in engorgement were evaluated.

**Inflammation at the bite site**. To determine the degree of inflammation at the bite site, skin samples were taken from mice infested with *I. scapularis* nymphs at day 3. Control samples consisted of skin from non-infested mice matching the location where ticks had bitten the infested mice. Biopsy punches of 2-mm were taken, and skin was trimmed to the proximity of the tick mouthparts. Skin samples were placed in 10% Neutral Buffered Formalin (v/v) for 2–3 weeks before parafilm embedding. Tissue was vertically cut, and sections were hematoxylin and eosin (H/E) stained. Pictures were taken with a Leica DFC495 microscope (Leica). The thickness of the epidermis was measured using ImageJ.

**Chemokine and cytokines**. *I. scapularis* nymphs were microinjected with *vamp33 si* or *vamp33 sc* RNAs. Microinjected ticks were placed on C57BL/6J (WT) mice and allowed to feed for 3 days before the skin biopsy. The bite site was marked with a permanent marker and ticks were removed with forceps. Skin biopsies were taken from the bite site using a 3-mm Integra® Miltex disposable biopsy punch (Integra). The weight of the punch biopsies was measured using a Pioneer™ analytical balance (OHAUS). Biopsies were placed in dry ice and into −80 ℃ for storage. Biopsies were homogenized in protein extraction buffer[95], containing 0.5% BSA (Sigma), 0.1% IGEPAL® CA-630 (Sigma), and 1% Halt™ protease inhibitor (Thermo Scientific), and normalized by weight (500 µl/mg of skin). Homogenized lysates were stored at −80 ℃ until used.

The relative concentration of the chemokines CXCL10, CXCL1, CXCL2, CCL2, CCL3, CCL4, and CCL20 were measured using the U-PLEX Chemokine Combo Mouse (Meso Scale Diagnostics) following the manufacturer's instructions. The concentration of Th1/Th2 (Interferon (IFN)-γ, Interleukin (IL)-10, IL-12p70, IL-13, IL-1β, IL-2, IL-4, IL-5 and Tumor Necrosis Factor (TNF)-α) and Th17 (IFN-γ, IL-17A, IL-17C, IL-17C, IL-17E/IL-25, IL-17F, IL-21, IL-22, and IL-6) cytokines were analyzed using the U-PLEX TH1/TH2 Combo Mouse and U-PLEX TH17 Combo 2 Mouse, respectively. Samples were measured and concentrations were analyzed with the MSD discovery workbench – methodical mind version 2013-2019 (Meso Scale Diagnostics).

**Skin immune cell populations**

*Cell populations affected during tick feeding*. *I. scapularis* nymphs fed on C57BL/6J mice. At the 3rd day of feeding, mice were euthanized with $CO_2$ and the site where

ticks were feeding was shaved. A 10-mm skin punch biopsy was taken while ticks were still attached. Skin samples from control mice were taken from matching locations. Single cell suspensions were prepared from each skin sample. Briefly, skin samples were cut into small pieces with sterile surgical scissors and placed into 14 ml FALCON® polypropylene round-bottom tubes containing 3 ml digestion buffer consisting of 90% RPMI-1640 (Quality Biological), 10% Liberase™ TL Research Grade (Roche), and 0.1% DNAse I (Millipore-Sigma). Digestions were carried out for 1 h and 5 min at 37 ℃ on constant shaking at 70 rpm. Single cell suspensions were obtained by passing the digested tissues through a 40-µm cell strainer (Corning), homogenizing the tissue with a plunger and flushing cells with 20 ml RPMI-1640. Cells were centrifuged at 568×g for 5 min at 4 ℃, resuspended in 1 ml FACS buffer (PBS containing 1% BSA). In total, 200 µl of the suspension were placed into a 96-well U-bottom plate and surface-stained with antibody panels.

Granulocytes (neutrophils, macrophages, monocytes, eosinophils, and basophils) were labeled with APC/Vio770 anti-CD11c (Miltenyi Biotec 130-107-461; 3 ŋg/µl), VioBlue anti-Ly-6G (Miltenyi Biotec 130-107-461; 3 ŋg/µl), FITC anti-Ly-6C (Miltenyi Biotec discontinued; 3 ŋg/µl), PE anti-Siglec-f (Miltenyi Biotec 130-102-274; 3 ŋg/µl), APC anti-CD11b (Miltenyi Biotec 130-113-793; 3 ŋg/µl), and PE/Vio770 anti-F4/80 (Miltenyi Biotec 130-118-459; 3 ŋg/µl). Mast cells were labeled with APC/Vio770 anti-CD45 (Miltenyi Biotec discontinued; 3 ŋg/µl), FITC anti-B220 (Miltenyi Biotec 130-102-810; 3 ŋg/µl), PE anti-CD11b (Miltenyi Biotec 130-113-235; 3 ŋg/µl), APC anti-CD117 (Miltenyi Biotec 130-102-796; 3 ŋg/µl), PE/Vio700 anti-CD3ε (Miltenyi Biotec 130-102-359; 3 ŋg/µl), and eFluor 450 anti-FceR1 (eBioscience 48589882; 2 ŋg/µl). Lymphocytes (T and B cells) were labeled with APC/Vio700 anti-CD8a (Miltenyi Biotec 130-102-305; 3 ŋg/µl), VioBlue anti-CD45 (Miltenyi Biotec 130-119-130; 3 ŋg/µl), FITC anti-TCRγ/δ (Miltenyi Biotec 130-104-015; 3 ŋg/µl), PE anti-CD4 (Miltenyi Biotec 130-102-619; 3 ŋg/µl), APC anti-B220 (Miltenyi Biotec 130-102-259; 3 ŋg/µl), and PE/Vio770 CD3 (Miltenyi Biotec 130-116-530; 3 ŋg/µl). Live and dead cells were discriminated using propidium iodide solution (Miltenyi Biotec). Cells were incubated with an antibody panel for 10 min at 4 ℃ and washed with 1% BSA with 2 mM EDTA. Fluorescently labeled populations were measured with a MACSQuant flow cytometer (Miltenyi Biotec) and analysis was performed using the FlowJo software v 10.6.1 (Treestar).

Cell types were defined by a flow cytometry gating strategy, as indicated in Supplementary Fig. 12. Myeloid cells were gated on the CD45+CD11b+ population from live cells. Neutrophils and monocytes were identified as Ly6CloLy6Ghi and Ly6ChiLy6Glo cells, respectively. Eosinophils were identified as CD11b+Siglec-F+ from the CD45+ population of live cells. Basophils were identified as CD11b+FceR1+ cells from the CD45+ population of live cells. Macrophages were identified as CD11b+F4/80int cells from the CD45+ population of live cells. DCs were identified as CD11b+CD11c+ cells from the CD45+ population of live cells. Mast cells were first gated on the B220−CD3− population from CD45+ cells followed by identification as FceR1+CD117+ cells. T cells were identified in live cells that were first gated on CD45+. γδ T cells, CD4+, and CD8+ T cells were identified as CD3+TCRγδ+, CD3+CD4+, and CD3+CD8+, respectively. B cells were gated on the B220+ population from CD45+ cells.

DETC populations in the murine skin were labeled with APC anti-CD45 (BioLegend 103111; 1:100), FITC anti-CD3 (BioLegend 100203; 1:100), BV60 anti-Vγ5 (BD 743241; 1:50), monoclonal antibody 17D1[96] (kindly provided by Dr. Adrian Hayday, King's College London, and Dr. Robert Tigelaar, Yale University), and PE mouse anti-rat IgM (BD 553888; 1:100). Live and dead cells were discriminated using Zombie Violet Fixable Live Dead stain (BioLegend). Cells were washed with FACS buffer (1% BSA in PBS with 2 mM EDTA and 0.05% NaN3). Cells were then blocked with anti-FcR (CD16-CD32) (BioLegend 156603; 1:500), and subsequently stained with the antibody panel for 15 min at 4 ℃ and washed with FACS buffer. Anti-rat IgM was added to the cells, incubated for 15 min at 4 ℃, and washed twice with FACS buffer. Finally, cells were resuspended in 4% paraformaldehyde, measured with an LSRII flow cytometer (BD) and analysis was performed using the FlowJo software v 10.6.1.

*Cells populations affected by EVs*. Exosome biogenesis was reduced in *I. scapularis* nymphs with *vamp33 si*. Control *I. scapularis* nymphs were microinjected with *vamp33 sc*. Nymphs fed for 3 days and skin samples were taken as described above.

**γδ T cell depletion**. C57BL/6J mice were injected intraperitoneally (i.p.) with 500 µg of InVivoMAb anti-mouse TCR γ/δ clone UC7-13D5 (BioCell BE0070) or InVivoMAb polyclonal Armenian hamster IgG (BioCell BE0091; control) on the day (day -1) and on the morning (day 0) prior to tick placement. Ticks were microinjected with *vamp33 si* or *vamp33 sc*. Ticks fed on γδ T cell-depleted and isotype control mice for 3 days. Mice were euthanized on the third day. Tick weights were measured. Depletion was confirmed through FACS measurement of γδ T cells in the skin and spleens. Skin samples were labeled using VioBlue anti-CD45 (Miltenyi Biotec 130-119-130; 3 ŋg/µl), FITC anti-TCRγ/δ (Miltenyi Biotec 130-104-015; 3 ŋg/µl), PE/Vio770 CD3 (Miltenyi Biotec 130-116-530; 3 ŋg/µl), and PE anti-TCR β Chain (BD 553172; 2 ŋg/µl). Live and dead cells were discriminated using propidium iodide solution (Miltenyi Biotec). T cell populations were measured with a MACSQuant flow cytometer and analysis was performed using FlowJo software v 10.6.1.

**Genetic ablation of αβ and γδ T Cells**. Ticks were microinjected with *vamp33 si* or *vamp33 sc* and placed on C57BL/6J (WT), *ccr2*$^{-/-}$, *ccr6*$^{-/-}$, *tcrδ*$^{-/-}$, *tcrβ*$^{-/-}$, FVB/N Jax, and FVB/N Tac mice. Ticks fed for 3 days and their weight was measured.

**FTY720 treatment in mice**. C57BL/6J (WT) mice were injected intraperitoneally (i.p.) with FTY720 (Sigma-Aldrich), 1 mg/kg in 100 μL sterile water on days −1 and on the morning of day 0 prior to tick placement.

Ticks were microinjected with *vamp33 si* or *vamp33 sc*. Ticks fed on FTY720-injected and PBS-injected mice for 3 days. Mice were euthanized on the third day. Tick weights were measured. In all, 10 mm skin punch biopsies and draining lymph nodes were obtained from euthanized mice. FTY720 treatment was confirmed through FACS measurement of lymphocytes in the skin and draining lymph nodes. Skin and lymph node samples were labeled using APC-Cy7 anti-CD45 (Miltenyi Biotec 130-110-662; 3 ng/μl), VioBlue anti-CD4 (Miltenyi Biotec 130-118-568; 3 ng/μl), PerCP-Vio700 anti-TCRγ/δ (Miltenyi Biotec 130-117-665; 3 ng/μl), and PECy7 CD3 (Miltenyi Biotec 130-116-530; 3 ng/μl). Live and dead cells were discriminated using propidium iodide solution (Miltenyi Biotec). Cell populations were measured with a MACSQuant flow cytometer and analysis was performed using FlowJo software v 10.6.1.

**Intracellular bacterium infection**

*A. phagocytophilum model*. Female C57BL/6 mice were shaved in three different spots on the back. Cell-free *A. phagocytophilum* ($1 \times 10^7$ bacteria/injection) resuspended in 50 μl of PBS were intradermally (i.d.) injected into these spots in the presence or absence of $4 \times 10^7$ *I. scapularis* nymphal salivary EVs. Infections progressed for 48 h and animals were euthanized using CO$_2$. Blood samples were taken by heart puncture and divided into two specimens: half of the blood in a serum collection tube and the other half in EDTA Microvette® 200 μl tubes (Sarstedt Ag). Skin biopsies were taken from each injection site, and spleens and livers were dissected. Skin and spleen samples were placed at −80 °C. DNA was extracted from skin samples, using the DNeasy Blood & Tissue Kit (QIAGEN) according to a modified version of a previously described protocol[97]. Briefly, the samples were grinded in liquid nitrogen (LN$_2$), using a mortar and pestle. Ground tissues were resuspended in 180 μl ATL Buffer and 20 μl of proteinase K. Tissues were incubated at 56 °C overnight. DNA purification was followed using the manufacturer's specification. DNA was eluted in 100 μl of elution buffer. DNA concentration was measured with a NanoDrop Spectrophotometer ND-1000 (Thermo Fisher Scientific).

Bacterial numbers at the skin site, blood, and other organs were measured by amplifying the single copy *16s* gene[98]. Gene expression was normalized by the mouse *actin* gene listed in Supplemental Data 6. Genes were amplified under the following conditions: a denaturation cycle of 10 mins at 95 °C, followed by 40 cycles of denaturalization for 15 s at 95 °C, amplification at two different annealing temperatures (*A. phagocytophilum 16s*: 48 °C and mouse *actin*: 55 °C) for 1 min. Bacterial quantification was performed with the Bio-Rad CFX96 Real-Time System through the CFX Maestro Software 1.1. The presence of a single peak was confirmed with the melting curve. Cytokines were measured as described above. Serum samples were centrifuged at 16,100×g for 5 min after 30 min incubation at room temperature. Serum samples were stored at −80 °C until use. Th1/Th2 cytokines were measured with the U-PLEX TH1/TH2 Combo Mouse, using a 1:2 dilution and following the manufacturer's recommendations.

*F. tularensis model*. *F. tularensis* live vaccine strain (LVS) was grown from glycerol stocks on Mueller Hinton Broth (MHB) plates (2.1% Mueller Hinton Broth (w/v; Difco), 0.5% sodium chloride (w/v; Fisher Scientific), 1% Tryptone (w/v; BD), 1.6% agar (w/v; BD), 0.1% glucose (w/v; Millipore-Sigma), 0.025% ferric pyrophosphate (w/v; Millipore-Sigma), 2% IsoVitalex (v/v; BD), and 10% sheep whole blood (v/v; Lampire) for 2 days at 37 °C with 5% CO$_2$. Bacteria were scrapped off and resuspended in MHB broth (2.1% Mueller Hinton Broth (w/v; Difco), calcium chloride 0.014% (w/v; Millipore-Sigma), magnesium chloride 1 mM (v/v; American Bioanalytical), 0.1% glucose (w/v; Millipore-Sigma), 0.025% ferric pyrophosphate (w/v; Millipore-Sigma), and 2% IsoVitalex (v/v; BD) for stocks at $18 \times 10^{11}$ CFU/ml that were frozen at −80 °C. Female mice were injected with bacteria at various concentrations to determine the correct lethal intradermal dose. The stock of bacteria was diluted to give $7.5 \times 10^6$, $7.5 \times 10^7$, and $7.5 \times 10^8$ CFU of bacteria in 50 μl of endotoxin-free PBS. The inoculum was confirmed by serial dilution plating onto MHB plates and allowed to grow for 2 days at 37 °C with 5% CO$_2$. Infected mice were monitored twice a day until they showed signs of distress at which point they were monitored every hour until all animals died. Weight loss of each animal was also measured daily. To establish the effect of EVs derived from the adult female *D. andersoni* salivary glands during the infection of *F. tularensis*, 10 female mice per group were i.d. injected with $8.4 \times 10^7$ CFU of bacteria in the presence or absence of $1 \times 10^8$ EVs, and weight loss and survival were monitored, as described above.

To determine how tick salivary vesicles affected bacterial numbers and the inflammatory response to *F. tularensis*, mice were i.d. injected with 6.7–6.8 × 10$^7$ CFU of bacteria in the presence or absence of EVs ($1 \times 10^8$/injection) derived from adult female *D. andersoni* salivary glands. Three independent experiments were performed and two out of three *F. tularensis* mouse infection challenges showed

similar results. Infection progressed for 5 days under daily monitoring. At day 5, animals were euthanized, and the weight of the animals was taken. Organs (spleens, livers, and lungs) were dissected and weighed. Blood samples were taken in Microvette® 200 μl tubes for serum (Sarstedt Ag). Spleens were weighed with a Pioneer™ analytical balance (OHAUS) and corrected by the weight of the mice to establish the degree of splenomegaly. Sections of organs were used to quantify the number of bacteria by homogenizing the organs in 500 μl of PBS inside 2.0 ml BeadBug™ tubes, prefilled with 1.5 mm Zirconium beads, triple-pure, high impact (Benchmark Scientific), with a benchmark BeadBug Microtube Homogenizer Model D1030(E) (Benchmark Scientific). Pieces of organs were homogenized at $400 \times 10$ speed for 90 s. Homogenates were serially diluted and 10 μl from the $1 \times 10^{-1}$ to $1 \times 10^{-5}$ dilutions were plated onto MHB plates, which were incubated for 2 days at 37 °C with 5% CO$_2$ before colonies were counted. CFUs of bacteria were normalized by the weight of the organs. Th1/Th2 cytokines in serum were analyzed using the U-PLEX Th1/Th2 Combo Mouse, using a 1:10 dilution for *Francisella*-challenged mice (with or without EVs) and 1:2 for the control mice (PBS alone or PBS and EVs).

**Tick acquisition and transmission of *F. tularensis***. One day prior to tick placement, C3H/HeN mice were anesthetized with a ketamine-xylazine sedative, an area ~2.5 cm in diameter between the shoulder blades was shaved with surgical clippers, and plastic chambers (top portion of 15-mL conical tubes) were adhered to shaved skin using Kamar adhesive. Mice were individually housed to prevent chamber removal by cage mates. The next day (day -5), mice were anesthetized, nymphal *D. variabilis* ticks were placed in each chamber, chambers were closed with a fine-mesh polyester fabric, and fabric was secured to each chamber using a rubber band. Double-sided tape was adhered to the inside upper rim of each cage bottom and cages were placed onto tack mats to prevent loss of any escaped ticks.

Three days after tick placement (day -2), mice were anesthetized, and intravenously infected with LVS. Bacterial inoculum was serially diluted and plated in quadruplet onto sMHA to confirm CFUs. Approximately 48 h after infection (day 0), mice were anesthetized, replete ticks were collected, and blood was harvested from infected mice by cardiac puncture for serial dilution in PBS and plating. To determine bacterial numbers in ticks at various time points after repletion (e.g., day 0, week 2, week 4, week 6, etc.), ticks were first surface-sterilized by placing into 30% H$_2$O$_2$ for five seconds, 70% ethanol for 5 s, washed with molecular grade water (Corning) for 5 seconds, then homogenized in RNAse-free disposable pellet pestle tubes (Fisher) containing 200 μL of sterile PBS. Tick homogenates were serially diluted in PBS and plated onto MHA containing 100 mg/L cycloheximide, 80,000 U/L polymyxin B, and 2.5 mg/L amphotericin B. Following 72 h of incubation, the number of colonies per plate were counted and CFU/ml (mouse blood) or CFU/tick were calculated.

Infection studies were performed essentially as described above, with the following modifications: LVS-infected adult ticks (14 weeks after their nymphal blood meal) were individually placed into chambers and allowed to feed to repletion on naïve mice. Repleted adult ticks were collected 8 days after tick placement and were either immediately homogenized for bacterial enumeration (day 8) or homogenized for bacterial enumeration when associated mice were moribund (days 14–18). To quantitate bacterial burdens in mouse blood and tissues during infection studies, mouse blood/tissues were either collected on day 8 (tick repletion) or when mice were moribund and humanely euthanized (days 14–18). Mice were anesthetized, blood was collected by cardiac puncture, mice were cervically dislocated, skin from the tick attachment site was harvested using a 8 mm biopsy punch (Accuderm), and lungs, livers, and spleens were aseptically harvested and transferred to sterile Whirlpack bags. Samples were homogenized, 25 μl of PBS/mg of tissue was added to each tissue, serially diluted, and dilutions were plated onto media.

**Ethics statement**. Blood collections were performed on healthy volunteers who provided informed consent. The protocol was approved by the Institutional Review Board (IRB# HP-00040025) of the University of Maryland School of Medicine and comply with the 21 CFR part 50. *I. ricinus* feeding experiments were performed in accordance with the Animal Protection Law of the Czech Republic (§17, Act No. 246/1992 Sb) and with the approval of the Akademie Věd České Republiky (approval no. 161/2010). All mouse experiments were carried out under the guidelines approved by the Institutional Biosafety (IBC#00002247) and Animal Care and Use committees (IACUC#0216015 and #0119012) at the University of Maryland School of Medicine and (IACUC#108672 and IBC#108665) at the University of Toledo College of Medicine and Life Sciences according to the National Institutes of Health (NIH) guidelines (Office of Laboratory Animal Welfare (OLAW) #A3200-01, A323-01, and A3270-1). Calf experiments were done in accordance with the guidelines approved by the IACUC at the University of Idaho (IACUC# 2017-59).

**Statistical analysis**. Statistical significance of each experiment was assessed with the unpaired one or two-tailed Student's *t* test. One-way or Two-way ANOVA post-hoc Sidak test for multiple comparisons and Two-way ANOVA followed by Fisher's least significance difference statistical tests were also used whenever appropriate. Survival curves were analyzed with the Kaplan–Meier followed by the

Log-rank (Mantel-Cox) test. We used GraphPad PRISM® (GraphPad Software version 9.1.0) for all statistical analyses. Outliers were detected by a Graphpad Quickcals program (https://www.graphpad.com/quickcalcs/Grubbs1.cfm).

**Reporting summary**. Further information on research design is available in the Nature Research Reporting Summary linked to this article.

## Data availability

The mass spectrometry proteomics data have been deposited to the ProteomeXchange Consortium via the PRIDE[99] partner repository with the dataset identifier PXD018779 under the project name "Label free proteomics profiling of nanovesicle isolated from cultured salivary glands isolated from partially fed adult female *Ixodes scapularis*" – Project https://doi.org/10.6019/PXD018779. All other data are available upon reasonable request. Raw data from all experiments are available as Source Data file. Source data are provided with this paper.

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

## Acknowledgements

We thank Jon Skare (Texas A&M University Health Science Center) for providing the B. burgdorferi B31 strain, clone MSK5; Dr. Adrian Hayday (King's College London) and Dr. Robert Tigelaar (Yale University) for the monoclonal antibody 17D1; Ulrike G. Munderloh (University of Minnesota) for supplying ISE6, DAE100, and AAE2 tick cells; Regina Harley (University of Maryland School of Medicine) for flow cytometry assistance; Anthony Kim, Aniket Wadajikar (University of Maryland School of Medicine) and Tek Lamichhane (University of Maryland College Park) for technical support with the EV characterization; Ru-Ching Hsia and Joseph Mauban (University of Maryland School of Medicine) for aiding in the microscopy analysis; J. Stephen Dumler (University of Maryland School of Medicine) for providing the E. chaffeensis Arkansas strain; Jonathan Oliver (University of Minnesota) for making available I. scapularis ticks; Gaelle Noel (University of Maryland School of Medicine) for technical assistance with the isolation of human peripheral blood mononuclear cells and differentiation to macrophages; Katharina Richard and Stefanie Vogel (University of Maryland School of Medicine) for providing the F. tularensis LVS strain; and Cristiana Cairo (University of Maryland School of Medicine) for insightful advice related to γδ T cells. This work was supported by grants from the National Institutes of Health (NIH) and the National Science Foundation (NSF) to J.H.F.P. (R01AI134696, R01AI116523 and R01 AI049424), J.H.F.P. and U.P. (P01AI138949), U.P. (R01AI116620), E.E.M.C. (F31AI138440), A.J.O.

(F31AI152215), E.M.B. (R01AI123129) and S.M.J. (R01HL141611 and 1750542). This article was prepared while Amanda D. Buskirk, Ph.D. was employed at the University of Maryland School of Medicine and Center for Vaccine Development and Global Health. The content is solely the responsibility of the authors and does not necessarily represent the official views of the NIH, NSF, FDA, the Department of Health and Human Services or the United States government.

## Author contributions

A.S.O.C. performed the research with contributions from X.W., L.M., N.K.A., H.L.H., D.K.S., B.G.T., E.E.M.C., S.L.F., L.R.B., P.S., A.D.B., K.M., L.L., C.C.C., A.J.O., K.L.M., B.E.H., D.E.S., and M.K. Reagents and advice were provided by C.B.M., J.G.V., M.F.P., M.B.S., U.P., S.M.J., J.F.H., L.M., G.A.S., E.M.B., M.L.L., and L.S. The project was developed by A.S.O.C. and supervised by J.H.F.P with input from L.S., M.K., E.M.B., and L.S.M. The paper was written by A.S.O.C. and J.H.F.P. with help from the other authors.

## Competing interests

L.S.M. is a full-time employee of Janssen Pharmaceuticals and holds Johnson & Johnson stock. L.S.M. performed all work at his prior affiliation at Johns Hopkins University School of Medicine and he has received prior grant support from AstraZeneca, Pfizer, Boehringer Ingelheim, Regeneron Pharmaceuticals, and Moderna Therapeutics. L.S.M. was also a paid consultant for Armirall, AstraZeneca, Moderna Therapeutics and Janssen Research and Development. L.S.M. was on the scientific advisory board of Integrated Biotherapeutics and is a shareholder of Noveome Biotherapeutics. All of these aforementioned companies are developing therapeutics against infections and/or inflammatory conditions. The remaining authors declare not competing interest.
