## [Peer Review File · Nature Communications]

Reviewers' comments:

Reviewer #1 (Remarks to the Author):

In the work presented in the manuscript "Nanovesicles Act as a Molecular Rheostat Controlling Virulence During Vector-Borne Microbial Transmission" the authors describe how they characterized the proteomic composition of nanovesicles secreted by adult *I. scapularis* glands by shotgun proteomics. The identification of proteins potentially capable of modulating host physiology encouraged the authors to orientate following experiments towards investigating whether the vector could either promote or mitigate pathogen-host interaction. This concept, which is the cornerstone of the manuscript, seems not fully supported by experimental data. For example, the general statement of the abstract: "We show that tick nanovesicles only promote pathogen dissemination when there is mostly a neutral relationship between the microbe and the mammal." is actually supported by only two observations concerning *F. tularensis* (lethal) and *A. phagocytophilum* (mildly virulent).

Though, as stated above, authors' conclusions are currently not supported by strong evidence, the manuscript and supplementary information are clearly written. The quality of the presentation, also concerning artwork is good. Methods are described with sufficient details.

The proteomics section of the manuscript rises a few questions:

- The objective of replicate experiments is to reinforce evidence. The authors would present more solid data by reporting proteins identified in both proteomic experiments instead of combining identifications of both replicates.
- Protein FDR (L534-L548): The authors speak about two independent data validations at the protein level (one in Peaks and the second one in Scaffold). It is not clear whether two lists have been combined and how. The protein FDR of the final list should be 1% or better (I would consider 1.2% acceptable, but not 5%).
- On P8, L141, the authors state that ALIX, TSG101 and CD63 were detected by mass spectrometry. The first two proteins, though, are not present in Supplementary Table 2.
- Supplementary Table 2 reports 579 proteins, whereas Supp Fig 2e Venn diagrams suggest that 350 proteins were identified.
- The authors state that nanovesicles revealed and overrepresentation of proteins connected to integrin signaling (L150), but this biological process does not seem to be enriched in bioinformatic analysis. Please clarify.
- L520: De novo sequencing is a specific term used in proteomics to indicate an attempt of sequencing without performing a database search. Its use in this context, even with quotation marks, is not correct.
- L489: probably it was the analytical column which was equilibrated with 3 microliters of 5% acetonitrile.

The data deposited online (on vesiclepedia and PRIDE) are currently not accessible to reviewers because the authors apparently did not provide temporary passwords to the Editor.

Reviewer #2 (Remarks to the Author):

A large team of collaborators have produced an enormous quantity of data. They have compiled an extensive list of components identified by proteomics in the extracellular vesicles. They demonstrated that complete feeding is dependent upon tick salivary gland nanovesicles. Complete feeding presumably mediated by extracellular vesicles secreted by the ticks was mediated by inhibition of Gamma Delta T cells, IL-5, interleukin 13, interleukin 2, CXCL10, CCL20, and CCL2.

Some of the concepts and experimental design require greater support. Tick saliva has its effect at the host- pathogen- tick interface. This is in the dermis of the skin and direct systemic effects are minimal. The introduction presents needle inoculation of salivary gland extracts as reflective of tick feeding. Ticks feed over many days with continuous secretion of tick saliva. The situation is very different from bolus inoculation of tick salivary gland extract. The concept that nanovesicles from *Dermacentor andersoni* ticks reduce spreading of the pathogen *Francisella* is not based upon the true scenario of transmission. Tick feeding is likely complete or near complete during the incubation period before the onset of illness with tularemia.

In addition to the above comments, the following specific comments should be addressed:

1. The variation in sizes of the nanovesicles in the graphs in the figures is much larger than depicted in the text, for example plus or -7, plus or -6 and plus or -3 nm on line 124, similarly on lines 135-136.
2. Do the cell culture -derived vesicles and the tick salivary gland -derived vesicles have identical proteomic content?
3. Were the nanovesicles obtained from infected ticks identical in content other than the oxidative stress effect compared with vesicles from uninfected ticks?
4. Line 279: Tularemia transmitted to humans by tick bite is usually not fatal.
5. The hypothesis stated on lines 281-282 is not proven in this study.
6. The increased secretion of the Nrf2-regulated protein 6-PGD is a result of the pathologic effect of *Anaplasma* on the tick and cannot be seen as a mechanism of manipulation of the tick to the arthropod's advantage.
7. Line 627: What was the origin of the PHK26-labeled nanovesicles, tick salivary glands or cell culture?
8. Lines 849-860: This experimental design does not reflect anything that would happen in nature. Mice inoculated intraperitoneally with LPS and nanovesicles do not reflect in any way tick feeding over a period of days with prolonged inoculation of tick saliva and pathogens into the skin.
9. Lines 903-925: This experiment is better designed than the intraperitoneal LPS experiment; however, does the inoculation of more than 10^7 colony forming units of *Francisella* mimic the quantity that would be inoculated during tick feeding? Does the inoculation of 10^8 nanovesicles reflect the quantity that would be injected during tick feeding? I suspect that the quantities of bacteria and nanovesicles in the bolus inoculation do not reflect the events in tick feeding.
10. Statistical analyses are missing for many of the quantitative comparisons that are reflected as meaningful.
11. Figure 6e demonstrates a difference in splenomegaly. It is important to recognize that the spleen size reflects congestion, B cell and T cell hyperplasia, and extramedullary hematopoiesis. These may reflect physiological responses. It is incorrect to refer to bacterial loads in organs as bacteremia. It is unclear what was measured in figure 6h.
12. Figure S3: How are the lengths of the bars calculated?

Reviewer #1 (Comments)

In the work presented in the manuscript “*Nanovesicles Act as a Molecular Rheostat Controlling Virulence During Vector-Borne Microbial Transmission*” the authors describe how they characterized the proteomic composition of extracellular vesicles secreted by adult *I. scapularis* glands by shotgun proteomics. The identification of proteins potentially capable of modulating host physiology encouraged the authors to orientate following experiments towards investigating whether the vector could either promote or mitigate pathogen-host interaction. This concept, which is the cornerstone of the manuscript, seems not fully supported by experimental data. For example, the general statement of the abstract: “We show that tick nanovesicles only promote pathogen dissemination when there is mostly a neutral relationship between the microbe and the mammal.” is actually supported by only two observations concerning *F. tularensis* (lethal) and *A. phagocytophilum* (mildly virulent). Though, as stated above, authors’ conclusions are currently not supported by strong evidence, the manuscript and supplementary information are clearly written. The quality of the presentation, also concerning artwork is good. Methods are described with enough details.

In the revised manuscript, we have made our statements more specific, particularly, when related to the technical limitations surrounding the tick models for *F. tularensis* (Lines 341-351). We included extensive new findings, which strengthened our arguments and improved the language and content of our manuscript. In this revised manuscript, we: (1) developed a tick salivary organoid system that mimics extracellular vesicle release by ticks; (2) described an extraction protocol for tick extracellular vesicles, which will be readily available to the scientific community worldwide; (3) characterized the proteomic composition of extracellular vesicles secreted by adult *I. scapularis* glands; (4) showed that extracellular vesicles derived from cultured cells and salivary glands had a different protein cargo and post-translational profile, and microbial infection affects some post-translational modifications of moieties inside extracellular vesicles; (5) developed a model of *D. variabilis* transmission of *F. tularensis* to mice and estimated the inoculation amount of this microbe during tick transmission to mice, which nicely correlated with their *bona fide* natural habitat; (6) expanded the current paradigm for ectoparasite feeding on mammals by showing that tick extracellular vesicles regulate dendritic epidermal T cells in the skin for an optimal feeding environment; and (7) proposed that extracellular vesicles act as a molecular rheostat dictating microbial virulence during vector-borne transmission. Collectively, the revised manuscript has considerably improved depth surrounding our claims.

The proteomics section of the manuscript rises a few questions:

1) The objective of replicate experiments is to reinforce evidence. The authors would present more solid data by reporting proteins identified in both proteomic experiments instead of combining identifications of both replicates.

In the revised manuscript, we deposited individual experiments in the ProteomeXchange Consortium via the PRIDE (Proteomics Identification Database) partner repository with the dataset identifier PXD013839 and 10.6019/PXD013839, as suggested by Reviewer #1. The protein list has also been shared in the Vesiclepedia database (www.microvesicles.org). We used distinct mass spectrometers for peptide separation and identification in both experiments. We highlighted this aspect of our analysis with an asterisk in Table S2. The first experiment was performed on an Orbitrap Velos mass spectrometer instrument with a few nanograms of starting material. For this analysis, we had to dissect hundreds of *Ixodes scapularis* specimens. This effort was what we considered technically possible for tick dissections in an academic laboratory. Despite these efforts, this experiment only retrieved 82 tick proteins in the tick extracellular vesicles. In early 2018, the authors had access to the superior and more sensitive mass Q Exactive HF-X Orbitrap mass spectrometer instrument (Sun et al., 2013 - *Rapid Commun Mass Spectrom.*). The Q Exactive HF-X Orbitrap instrument retrieved 321 proteins from tick extracellular vesicles with the same amount of material. The figure below indicates the number of proteins that were shared between the two experiments. We noted the use of different mass spectrometry instruments for both experiments in the Materials and Methods (Lines 598-600). Importantly, we validated protein hits from both mass spectrometry instruments through western blots (Figure 1h).

2) Protein FDR (L534-L548): The authors speak about two independent data validations at the protein level (one in Peaks and the second one in Scaffold). It is not clear whether two lists have been combined and how. The protein FDR of the final list should be 1% or better (I would consider 1.2% acceptable, but not 5%).

We used both PEAKS and Scaffold to perform the MS/MS analysis, as described in the Materials and Methods (Lines 635-663). Table S2 presents the protein identification from two independent experiments. Specifically, 389 out of 579 proteins have FDR<1.2% for peptides and FDR <0.8% for proteins as analyzed by Scaffold and the built-in Prophet algorithm. Additional proteins were exported from PEAKS with an FDR between 1.2% and 5%; these proteins were checked manually for their MS/MS (fragmentation profiles) and presence of unique peptides. Upon validation, these proteins were added to Table S2. From our experience using PEAKS, we observed that exporting proteins with FDR<5% helps improve the proteomic output, particularly in experimental condition, such as the one reported here, where the amount of material is scarce. However, to ensure correct peptide/protein assignment all the PEAKS imported data were re-evaluated with Scaffold and manually.

A pertinent issue here is the technical difficulty for obtaining material from the saliva of *I. scapularis*. We had to dissect approximately 500 *I. scapularis* specimens to obtain just a few hundred nanograms of tick extracellular vesicles. Hence, we decided to report the tryptic peptides and assigned protein IDs with an FDR between 1.0-5.0% after manual inspection/validation of MS/MS profiles. This result is at the technical edge of what is currently possible in the field of proteomics for non-model organisms, such as *I. scapularis* ticks.

3) On P8, L141, the authors state that ALIX, TSG101 and CD63 were detected by mass spectrometry. The first two proteins, though, are not present in Supplementary Table 2.

The manuscript has been amended on this point (Lines 146-147). We wrote: “The presence of the exosomal markers ALIX, TSG101 and CD63 was detected in tick salivary EVs by immunoblots (Figs. 1e-g), as shown previously.”

4) Supplementary Table 2 reports 579 proteins, whereas Supp Fig 2e Venn diagrams suggest that 350 proteins were identified.

Venn diagrams represented a comparison between proteins present in our dataset and those that have been deposited in Vesiclepedia. The Vesiclepedia database typically displays proteins deposited by the scientific community in vesicle populations from model organisms (e.g., humans, mice, among others). Therefore, our results with *I. scapularis*, a non-model organism, will never achieve 100% match with any database because of the lack of representation for tick proteins. For the revised manuscript, we have re-done our analysis (April 2020) (Supplementary Figure 2e). Please refer to the Materials and Methods for more details.

5)The authors state that nanovesicles revealed and overrepresentation of proteins connected to integrin signaling (L150), but this biological process does not seem to be enriched in bioinformatic analysis. Please clarify.

We re-checked our analysis at the request of Reviewer #1. The overrepresentation of the integrin signaling pathway was detected by the ingenuity pathway analysis (IPA) software

(<https://digitalinsights.qiagen.com/>) (Figure 2A). All proteins highlighted in Figure 2B, which are part of the integrin transduction machinery, were also reported in Table S2.

6) L520: *De novo* sequencing is a specific term used in proteomics to indicate an attempt of sequencing without performing a database search. Its use in this context, even with quotation marks, is not correct.

In the PEAKS algorithm, *de novo* sequencing is performed by default before any database search. *De novo* sequenced data was assigned for protein identification against *I. scapularis* in the SwissProt database (January 2018; 21974 entries).

7) L489: probably it was the analytical column which was equilibrated with 3 microliters of 5% acetonitrile.

We equilibrated the peptide solution and the analytical column before separation. We used only a small volume of 3 microliter containing 5% acetonitrile and 0.1% formic acid to equilibrate the peptide solution before injection. Lines 604-608 were rewritten to avoid any confusion.

8) The data deposited online (on vesiclepedia and PRIDE) are currently not accessible to reviewers because the authors apparently did not provide temporary passwords to the Editor.

The reviewers should be able to retrieve all datasets related to this project from the PRIDE database on behalf of *Nature Communications*. Please refer to the PRIDE database guidelines for reviewers (<https://www.ebi.ac.uk/pride/help/archive/reviewers>). In short, the editor of *Nature Communications* must first contact the PRIDE database for data analysis. Then, reviewers need to be logged-in on behalf of the journal in the PRIDE system to view the data during the peer review process. After the journal contacts the PRIDE database with the PX number, the reviewer account will be created. We are sharing with the journal editors our account upon the manuscript submission. Please see below our information:

Project Title: "Label-Free Proteomics Profiling of Nanovesicles Isolated from Cultured Salivary Glands from Partially-Fed Adult Female *Ixodes scapularis*"

Project Accession Number: PXD013756 and DOI: 10.6019/PXD013756.

Username: reviewer74856@ebi.ac.uk

Password: gd80wTLm.

In the event the dataset does not become available to Reviewer #1, we will be pleased to provide the RAW files directly to the handling editor of *Nature Communications* for peer-review.

Reviewer #2 (Comments)

A large team of collaborators have produced an enormous quantity of data. They have compiled an extensive list of components identified by proteomics in the extracellular vesicles. They demonstrated that complete feeding is dependent upon tick salivary gland nanovesicles. Complete feeding presumably mediated by extracellular vesicles secreted by the ticks was mediated by inhibition of Gamma Delta T cells, IL-5, interleukin 13, interleukin 2, CXCL10, CCL20, and CCL2. Some of the concepts and experimental design require greater support:

- 1) Tick saliva has its effect at the host-pathogen-tick interface. This is in the dermis of the skin and direct systemic effects are minimal.

Indeed, most studies in the scientific literature have portrayed the effects of the tick bite as taking place within the feeding cavity or close to its vicinity (local effect). However, the skin is not a separate entity from other organs and immunological cues that are presumably initiated at the skin site can be propagated to distant cells and cause a systemic inflammatory response (Belkaid and Tamoutounour, 2016 - *Nature Reviews Immunology*; Chen *et al.*, 2018 - *Nature*). Additionally, extracellular vesicles can disseminate systemically (Hoshino *et al.*, 2015 - *Nature*; Poggio *et al.*, 2019 - *Cell*). An earlier report using immunofluorescence examination of where salivary antigens may be trapped in the skin

revealed that salivary proteins may affect an area beyond the dermal-epidermal location, suggesting a more widespread impact of these molecules within the mammalian host (Allen *et al.*, 1979 – *Immunology*).

Our findings indicate that extracellular vesicles from *D. andersoni* protects against *F. tularensis*-induced systemic inflammation (Lines 296-308; Figure 7a-l; Supplementary Figure 11). Therefore, although *F. tularensis* causes disease through systemic inflammation, skin mediators may be released into the blood stream to initiate a multi-organ response. Hence, we can reasonably deduct that tick extracellular vesicles act locally, as correctly pointed by Reviewer #2, but it also has distal and systemic effects (Lines 372-382).

Thus far the effect of the tick bite has been recorded in the literature as localized to the dermis. However, saliva may be injected in the epidermis, dermis or the epidermis/dermis border depending on the tick mouthpart apparatus (Wikel, 2013 – *Frontiers in Microbiology*). The epidermis contains numerous immune cells that may take up salivary antigens and transport them to draining lymph nodes (Bernard *et al.*, 2020 – *Trends in Parasitology*; Kobayashi *et al.*, 2020 – *Trends in Immunology*). In our revised manuscript, we demonstrated the effect of tick extracellular vesicles in DETCs, which are located in the mouse epidermis and are important for wound healing (Jameson *et al.*, 2002 – *Science*; Jameson and Havran, 2007 - *Immunol Rev.* 2007; Strid *et al.*, 2009 - *Semin Immunol.*; Havran and Jameson, 2010 – *J. Immunol.*; Komori *et al.*, 2011 – *J. Immunol.*). We performed a series of complex experiments that reveal causation in our findings to support our conclusions (Figures 4d-h; Supplementary Figure 9). First, we placed ticks injected with *vamp33* siRNA on mice with or without the FTY720 treatment. FTY720 inhibits lymphocyte egress from lymph nodes (Chiba *et al.*, 1998 - *J. Immunol.*; Brinkmann *et al.*, 2002 - *J. Biol. Chem.*; and Mandala *et al.*, 2002 - *Science*). This molecule allowed our group to determine whether the effect of tick extracellular vesicles was on resident or infiltrating $\gamma\delta$ T cells. We measured tick weight at day 3 post-attachment and detected an increase in the number of $\gamma\delta$ T cells at the mammalian skin site, as well as a 64% reduction in the weight of nanovesicle-deficient ticks placed on FTY720 treated mice. These findings suggested that the effect of tick extracellular vesicles during feeding was occurring at the resident, but not on infiltrating $\gamma\delta$ T cells (Figures 4d-e).

Second, we determined whether the effect of tick extracellular vesicles was on skin resident epidermal or dermal $\gamma\delta$ T cells. Murine skin resident $\gamma\delta$ T cells include dermal $\gamma\delta$ T cells (also known as $\gamma\delta$ T17 cells) that express the T cell receptor V γ 4 or V γ 6 (Gray *et al.*, 2011 - *J Immunol.*) and epidermal DETCs that express the T cell receptor V γ 5V δ 1 (Jameson and Havran, 2007 - *Immunol Rev.* 2007; and Strid *et al.*, 2009 - *Semin Immunol.*). Dermal $\gamma\delta$ T cells rely on the chemokine receptors CCR2 and CCR6 for cell recruitment to inflammatory sites and CCR2 for homeostatic trafficking (McKenzie *et al.*, 2017 - *Nat Commun.* 2017). We placed ticks injected with *vamp33* siRNA on *ccr2*^{-/-} and *ccr6*^{-/-} mice to determine if dermal $\gamma\delta$ T cells were being affected by tick extracellular vesicles. Genetic ablation of CCR2 and CCR6 did not affect extracellular vesicle-dependent tick feeding. These findings suggested that the dermal $\gamma\delta$ T cells were not involved in tick feeding (Supplementary Figure 9).

Third, we evaluated the skin DETC (V γ 5⁺V δ 1⁺) T cell population (Havran *et al.*, 1989 - *PNAS*; Mallick-Wood *et al.*, 1998 – *Science*; and Roarke *et al.* 2004 - *J Leukoc Biol.*) at the local tick bite site to assess whether tick extracellular vesicles regulated epidermal $\gamma\delta$ T cells. The majority of $\gamma\delta$ T cells observed in earlier experiments were in the TCR $\gamma\delta$ ^{high} group, which are the DETCs cells (Figure 4g). We placed ticks injected with *vamp33* siRNA on wildtype mice and measured tick weight at day 3 post-attachment. We detected a significant increase in frequency of epidermal $\gamma\delta$ T cells at the respective mammalian skin site from the *vamp33* siRNA treatment (Figure 4f). Next, we investigated the effect of DETCs on tick feeding by placing ticks injected with *vamp33* siRNA on FVB-Tac mice to provide causation to these findings. FVB-Tac mice are naturally depleted of DETCs due to a failure of thymic selection because of a natural mutation of the *Skint1* gene (Lewis *et al.*, 2006 - *Nat. Immunol.*; and Boyden *et al.* 2008 - *Nat. Genet.*; Barbee *et al.*, 2011 – *PNAS*; Turchinovich and Hayday, 2011 - *Immunity*). A decrease in weight was not observed when extracellular vesicle deficient ticks were placed on FVB-Tac (DETC

depleted) mice. Conversely, we detected the opposite findings in our experimental control group, which does not bear a mutation in the *Skint1* gene and harbored normal levels of DETCs (Figure 4h). Taken together, we discovered that tick extracellular vesicles regulate epidermal DETCs for an optimal feeding environment. This is a significant expansion of the previous paradigm established in the scientific community that tick saliva had its effect solely on the skin dermis.

- 2) The introduction presents needle inoculation of salivary gland extracts as reflective of tick feeding. Ticks feed over many days with continuous secretion of tick saliva. The situation is very different from bolus inoculation of tick salivary gland extract.

We have revisited the citations and corrected inaccurate references that present needle inoculation of salivary gland extracts as reflective of tick feeding. Unfortunately, it would be technically impossible to perform all experiments in our manuscript by only using tick saliva. It is necessary to obtain hundreds, if not thousands, of *I. scapularis* specimens to obtain extracellular vesicles from saliva for a given experiment. Due to this technical limitation, we adopted a hybrid isolation model where we only substantiated our results with EVs from saliva in the castor bean tick *I. ricinus*, the main vector of Lyme disease in Europe (Fig. 1I). A similar approach was also recently used in an unrelated article where the authors use extracellular vesicles from *I. scapularis* and *Amblyomma maculatum* to determine their effect on the HaCaT human keratinocyte cell line (Zhou *et al.*, 2020, *Frontiers in Cell and Developmental Biology*). In EVs derived from the saliva of *I. ricinus*, we identified the anti-inflammatory tick protein Sialostatin L2 and the tetraspanin CD63 in extracellular vesicles derived from adult saliva (Fig. 1I). We accommodated the thoughts of Reviewer #2 by having a paragraph in our article discussing technical limitations of our findings (Lines 154-163).

There is no perfect experiment that properly describes (in absolute terms) the process of tick saliva secretion in the feeding cavity. All animal models used to explore ectoparasite salivary secretion over the past few decades have their limitations. For instance, salivation of semi- or fully-engorged ticks with an application of a neuromodulator (e.g., pilocarpine, dopamine, norepinephrine, among others) has the disadvantage of artificially inducing tick salivation (Kotál *et al.*, 2015 – *J. Proteomics*; Simo *et al.*, 2017 – *Frontiers in Cellular and Infection Microbiology*). Conversely, the use of tick salivary gland extracts to phenocopy their effects on the feeding cavity may also not mimic the reality occurring in nature.

- 3) The concept that nanovesicles from *Dermacentor andersoni* ticks reduce spreading of the pathogen *Francisella* is not based upon the true scenario of transmission. Tick feeding is likely complete or near complete during the incubation period before the onset of illness with tularemia.

Whether tick feeding is complete or near complete during the incubation period before the onset of illness with tularemia remains a matter of controversy for almost 100 years. Therefore, it would be too bold for these authors to answer a century-old question with a single article. Parker and colleagues first isolated *Bacterium tularense* (now *Francisella tularensis*) from the Rocky Mountain Spotted Fever tick vector, *D. andersoni* (Parker *et al.*, 1924 – Public Health Reports). Subsequently, Green isolated the same bacterium from the American dog tick, *D. andersoni*, formally implicating ticks as a tularemia risk to humans (Green, 1931 – *American Journal Epidemiology*). Since then, researchers have debated the temporal kinetics of tick transmission of *Francisella* and the onset of tularemia in animal models with no avail (Zellner and Huntley, 2019 – *Frontiers in Cellular and Infection Microbiology*; Telford and Goethert, 2020 - *Annual Review of Entomology*). Over many decades, independent groups have used different tick infection techniques (e.g., feeding on infected animals versus capillary tube feeding), environmental versus laboratory experiments, animal models (e.g., mice, guinea pigs, rabbits, dogs), tick species (e.g., *D. variabilis*, *D. andersoni*, *Amblyomma americanum*), developmental stages (e.g., larvae, nymphs or adults) and *Francisella* genotype/strains (e.g., A1a, A1b, A2, B, LVS) to clarify this question. Taking these scientific constraints into consideration, our approach was to revise our manuscript and rework the language throughout the text to discuss limitations of our findings (Lines 341-351).

In addition to the above comments, the following specific comments should be addressed:

a) The variation in sizes of the nanovesicles in the graphs in the figures is much larger than depicted in the text, for example plus or -7, plus or -6 and plus or -3 nm on line 124, similarly on lines 135-136.

The means presented in the text are based on the average of the size mean from the Nanosight reports from all the runs, which takes into consideration the frequency of a specific vesicle population (Lines 572-576). Additionally, similar language was used throughout the text to avoid any confusion.

b) Do the cell culture-derived vesicles and the tick salivary gland-derived vesicles have identical proteomic content?

In the revised manuscript, we performed experiments with a subset of proteins detected in our proteomics analysis (Figure 1h). Some proteins were present in all EVs surveyed (e.g., DHX16 and PLS3), while others were detected only in EVs originated from tick cells (e.g., CTNNB1 and ABCC1) when compared to salivary glands (e.g., CCT7 and GFPT1). A similar pattern emerged for glycosylation in tick EVs (Figure 1i), but this observation was less pronounced for phosphorylation (Figure 1j) and carbonylation (Figure 1k).

c) Were the nanovesicles obtained from infected ticks identical in content other than the oxidative stress effect compared with vesicles from uninfected ticks?

In the revised manuscript, we performed these experiments. We only observed this effect for oxidative stress (Figure 5d). A similar effect was not observed for glycosylation (Figure 5f) or phosphorylation (Figure 5g).

d) Line 279: Tularemia transmitted to humans by tick bite is usually not fatal.

We have amended this sentence in the revised text to say that ulceroglandular tularemia, the most common type of disease caused by tick-transmission of *Francisella*, is moderate when compared to other tularemia forms (Lines 341-351). These clinical and epidemiological features agree with our findings where we show that tick extracellular vesicles are anti-inflammatory and mitigate the deleterious effects of *Francisella* in the mammalian host.

e) The hypothesis stated on lines 281-282 is not proven in this study.

We have eliminated unnecessary language from the revised manuscript.

f) The increased secretion of the Nrf2-regulated protein 6-PGD is a result of the pathologic effect of *Anaplasma* on the tick and cannot be seen as a mechanism of manipulation of the tick to the arthropod's advantage.

We have modified the text in the discussion and now reads “Whether the increase of 6-PGD results from the pathogenic or mutualistic effect of *A. phagocytophilum* remains to be determined.” (Lines 413-414).

g) Line 627: What was the origin of the PHK26-labeled nanovesicles, tick salivary glands or cell culture?

The origin was a tick cell line. We have clarified this in the text (Lines 765-775).

h) Lines 849-860: This experimental design does not reflect anything that would happen in nature. Mice inoculated intraperitoneally with LPS and nanovesicles do not reflect in any way tick feeding over a period of days with prolonged inoculation of tick saliva and pathogens into the skin.

We did not mean to say that mice inoculated intraperitoneally with LPS were reflective of the biology related to tick feeding. We corrected this language in the revised text to avoid any confusion (Lines 283-287). The goal of this experimental design was to investigate the anti-inflammatory effect of tick extracellular vesicles during systemic infection, as tick salivary effectors are known to have anti-inflammatory properties. Several studies conducted in the scientific literature have indicated that the LPS model leads to a strong inflammatory effect in mice, which

can be used as a read-out to evaluate acute inflammation. Therefore, we judged that this approach would be a good option to test our hypothesis.

i) Lines 903-925: This experiment is better designed than the intraperitoneal LPS experiment; however, does the inoculation of more than 10^7 colony forming units of *Francisella* mimic the quantity that would be inoculated during tick feeding? Does the inoculation of 10^8 nanovesicles reflect the quantity that would be injected during tick feeding? I suspect that the quantities of bacteria and nanovesicles in the bolus inoculation do not reflect the events in tick feeding.

To determine whether 1×10^7 CFU of *Francisella* used in our original experimental design mimicked the quantity of infection by tick transmission in nature, we first searched through the ecology and epidemiology literature. Surprisingly, *Dermacentor* ticks can harbor a very large amount of *F. tularensis* on Martha's Vineyard (median of 3.3×10^8 genome equivalent per tick; range from 0 to 10^{11} genome equivalent per tick) (Goethert and Telford, 2010 – *Ticks and Tick Borne Diseases*). Martha's Vineyard is an island off the coast of Massachusetts where epizootic transmission of tularemia by ticks has occurred since 2001 (Feldman *et al.*, 2001 – *New England Journal of Medicine*; Matyas *et al.*, 2007 – *Ann. NY Acad Sci*). Additionally, an older report of *Francisella* transmission by *Amblyomma americanum* provided a similar account of bacteria in ticks (mean of 6.7×10^7 CFU bacterium per nymphal ticks) (Hopla, 1960 – *Southern Med. J.*).

Nevertheless, the answer to this question remains exceedingly difficult because different tick species (e.g., *D. variabilis*, *D. andersoni*, *A. americanum*), developmental stages (e.g., larvae, nymphs or adults) and/or *Francisella* genotype/strains (e.g., A1a, A1b, A2, B, LVS) may affect the quantity of infection by tick transmission. Thus, we developed a model of *D. variabilis* transmission of *F. tularensis* to mice. We estimated the inoculation amount of this microbe during tick acquisition, persistence and transmission (Figures 7j-k). This model of tick infection considered the *D. variabilis* 5-day blood meal and the rapid lethality of *F. tularensis*. We observed that *F. tularensis* replicated inside ticks with an average of 9.5×10^5 CFU/tick after repletion, to a high of 6.5×10^6 CFU/tick of *F. tularensis* on week 4 post-feeding (Figure 7j). Subsequently, we assessed *D. variabilis* transmission of *Francisella* in the laboratory onto naïve mice. Examination of mouse blood revealed that 100% of naïve mice were infected with *F. tularensis* on day 14 after tick placement (6 days after tick repletion). Blood, skin, lungs, livers, and spleens were collected and *F. tularensis* was quantitated. We observed high bacterial burdens in blood (6×10^5 CFU/mL), lungs, livers, and spleens (3×10^5 – 1.9×10^6 CFU/mg tissue) (Figure 7k). Importantly, our reports corroborated with another article from an independent group where they developed a reproducible and quantitative model of *Francisella* in *D. variabilis* (Coburn *et al.*, 2015 – *Applied and Environmental Microbiology*). Despite the technical limitations of our system, we concluded that 1×10^7 CFU of *Francisella* in our experiments could be a realistic quantity of bacteria for a physiological system.

To the second question: “Whether the inoculation of 1×10^8 nanovesicles reflect the quantity that would be injected during tick feeding”. Unfortunately, the scientific community has not yet firmly developed technical approaches to visualize the effect of extracellular vesicles *in vivo*. This is major impediment in the field of vesicle biology and has been recently highlighted by several review articles (Robbins and Morelli, 2014 – *Nature Reviews Immunology*; Tkach and Thery, 2016 – *Cell*; van Niel *et al.*, 2018 – *Nature Reviews Molecular Cell Biology*; Kalluri and LeBleu, 2020 – *Science*). To date, only a few laboratories around the world have developed fluorescent proteins that are tagged to the plasma membrane to visualize extracellular vesicles *in vivo* by intravital microscopy or other sophisticated imaging techniques (Lai *et al.*, 2015 – *Nature Communications*; Verweij *et al.*, 2019 – *Developmental Cell*). This approach has limitations and it is difficult to visualize vesicles even in the context of super resolution microscopy (van Niel *et al.*, 2018 – *Nature Reviews Molecular Cell Biology*; Kalluri and LeBleu, 2020 – *Science*). Other groups have used transgenic mice expressing the CRE recombinase and a LacZ reporter gene to observe the effect of extracellular vesicles in animals (Ridder *et al.*, 2014 – *PLoS Biology*; Ridder *et al.*, 2015 –

Oncoimmunology; Zomer *et al.*, 2015 - *Cell*) or a CRISPR-based reporter system for single-cell detection of extracellular vesicles (de Jong *et al.*, 2020 – *Nature Communications*). Regrettably, these experimental approaches are not viable in ticks because they have a life cycle of two years (Eisen and Eisen, 2018 – *Trends in Parasitology*) and there are not any genome editing tools for *I. scapularis* (de la Fuente, 2018 – *Ticks and Tick-Borne Diseases*).

Given these technical impediments, we performed a literature search to determine what would be a reasonable range of extracellular vesicles used by the scientific community during experimentation. A cursory review of the literature revealed a number of experimental designs with extracellular vesicles ranging from 1×10^5 to 1×10^9 molecules (Zamanian *et al.*, 2015 – *PLoS Neglected Tropical Diseases*; Szempruch *et al.*, 2016 – *Cell*; Sisqueira *et al.*, 2017 – *Nature Communications*; Zhou *et al.*, 2018 – *PLoS Pathogens*; Vora *et al.*, 2018, *PNAS*; Sung *et al.*, 2019 – *Nature Communications*). Importantly, no severe immune reaction was observed in mice repeatedly injected with extracellular vesicle within this range for extended periods of time (Kamerkar *et al.*, 2017 – *Nature*; Zhu *et al.*, 2017 – *J. Extracellular Vesicles*; Mendt *et al.*, 2018 – *J. Clin. Investigation*). Therefore, we concluded that the injection of 1×10^8 extracellular vesicles in mice would be a judicious approach.

j) Statistical analyses are missing for many of the quantitative comparisons that are reflected as meaningful.

In the revised manuscript, we added the statistical analysis throughout the figures and legends. We also presented the use of statistics in the methods section (Lines 1125-1132). As per Nature Publishing Group guidelines, we also provide: (1) the comprehensive reporting summary and checklist of our manuscript, including all raw data (source data files); (2) detailed ethics statements; (3) statistical analyses; and (4) reagent availability and information in terms of how they were used in our manuscript (Table S4).

k) Figure 6e demonstrates a difference in splenomegaly. It is important to recognize that the spleen size reflects congestion, B cell and T cell hyperplasia, and extramedullary hematopoiesis. These may reflect physiological responses. It is incorrect to refer to bacterial loads in organs as bacteremia. It is unclear what was measured in figure 6h.

We have corrected this statement and eliminated the word “bacteremia” from the text.

l) Figure S3: How are the lengths of the bars calculated?

We first determined the most relevant pathways that were enriched in our proteomics dataset through the IPA software (<https://digitalinsights.qiagen.com/>). The IPA software is a commercial bioinformatics tool widely used by the scientific community that analyzes protein patterns using a build-in scientific literature database. Our analysis was done through default parameters and overrepresented pathways were listed on the y axis. The x axis indicated the correspondent Log (*p* value) for the IPA analysis. The Log (*p* value) analysis is done through a Qiagen proprietary algorithm. The higher the number on the x axis [Log (*p* value)] the more proteins from the dataset are fitted to the pathway. Log (*p* value) >1 indicates that a given pathway is overrepresented in the dataset (threshold). Log (*p* value) and threshold were indicated in the bottom of Figure S3 with a dotted and a thick continuous line, respectively.

In closing, we thank *Nature Communications* for the manuscript evaluation. We are excited that the reviewers found this study novel and of broad appeal for the scientific community.

Sincerely,

Joao Pedra, PhD
Associate Professor of Microbiology and Immunology

Reviewers' comments:

Reviewer #1 (Remarks to the Author):

The authors have addressed many points which, in my opinion, were unclear to some degree in the first version of the manuscript.

Nevertheless, some important information is still missing in the presentation of the proteomics data.

Before going to the point, I would like to remark that this time I was able to access proteomics data available on the PRIDE database. Whereas Q-Exactive data looked ok to me (I was able to spot 4 proteins identified by a single peptide which were probably false positives, but on over 300 identifications, this is more or less within expected FDR), even though I could not explain why more than 40 proteins were DECOY hits, on a total of less than 400 proteins, Velos data were not of the same quality. This is of course due to the much higher sensitivity and mass accuracy of Q-Exactive HF. Nevertheless, Velos data are anyway reported in Table S2 despite the fact that protein identifications are sensibly less convincing. In my opinion, many identifications obtained in the first analysis are false (bad MS/MS data, very few peaks correctly assigned to b and y ions). It has to be stressed that the first analysis identified very few peptides, thus the FDR estimation algorithm might not have had sufficient data to perform correctly. Furthermore, it is significant to notice that half of Velos identifications are not confirmed in the Q-Exactive analysis which, having identified 4x more proteins than Velos, should have identified the vast majority of proteins detected in the first analysis. This observation is also suggesting that many false identifications are present in the Velos data.

As the authors state, responding to my question number 2 (previous review):

"Additional proteins were exported from PEAKS with an FDR between 1.2% and 5%; these proteins were checked manually for their MS/MS (fragmentation profiles) and presence of unique peptides. Upon validation, these proteins were added to Table S2. From our experience using PEAKS, we observed that exporting proteins with FDR<5% helps improve the proteomic output, particularly in experimental condition, such as the one reported here, where the amount of material is scarce. However, to ensure correct peptide/protein assignment all the PEAKS imported data were re-evaluated with Scaffold and manually."

The information on which protein was validated and added manually should be added to Table S2. The statement that many proteins were identified by manual validation is apparently not present in the manuscript. Please add this information to the Methods section.

Besides, manual validation is a subjective procedure. The authors should present the validated and annotated MS/MS spectra as a separate file easily accessible in Supporting information in order to allow other researchers to assess the quality of the spectra. Needless to say, the MS/MS spectrum should strengthen the evidence for a poorly identified protein (just 1 peptide), thus it should be a spectrum which the majority of experts would consider of acceptable quality (unfortunately this is also a subjective estimation). The criteria used for manual validation should be listed in the Methods section of the manuscript (presence of justifiable missed cleavages, abundant proline-directed fragments, series

of at least 3 or at least 4 b or y ions, etc).

Please find a list of observations which, in my opinion, need to be addressed in order to strengthen the presentation of proteomics data:

- Table S2. The following columns should be added: number of identified peptides, number of unique peptides, whether the protein was identified by manual validation of a MS/MS spectrum, whether the identification was from QExactive or Velos.
- The Materials section should mention that manual validation was performed and should list the criteria adopted for manual validation.
- MS/MS spectra of manually validated identifications should be reported in Supporting information, together with annotation of b and y ions.
- Please update the correct PDX code in the manuscript (PXD013756)

I do acknowledge that excluding 30-50 proteins from a list might have important consequences on downstream bioinformatics, causing a “chain reaction” of modifications which might have unpredictable consequences. Nevertheless, I do believe that, unfortunately, proteomics data present in this manuscript need to have a more conservative evaluation before being published.

Reviewer #2 (Remarks to the Author):

The extensive responses to my queries and critiques address the issues that I have raised. Experimental design in this area is extremely difficult to achieve quantitative input that reflects natural tick feeding. The authors have done their best to justify their experimental designs.

The following comments are addressed on behalf of Reviewer#3, who was unavailable to re-review:

This manuscript represents a tremendous amount of high-quality experimental data. If the authors would tone down their interpretation and explain the weaknesses in their interpretations and data, the manuscript would merit publication in a journal of as high quality as Nature Communications. This is a complex study involving advanced methods, e.g., tick salivary organoids and proteomics. These data are very broad and yielded only moderately overlapping results. Indeed the study involved proteolytic analysis of tick salivary extracellular vesicles using two sources that yielded rather different results. Study of the effect on dendritic epidermal cells was more focused. The authors need to address the drawbacks of the data. The numerous references that were provided to address critique of reviewer #2 included a general review with no data on ticks and data from unrelated tumor cells. The authors have failed to effectively address the hypothesis that extreme dilution of salivary gland extracellular vesicles after leaving the dermis would make systemic effects homeopathic. Furthermore, the relevance of the $10^{<7}$ Francisella tularensis in a whole tick to its concentration in secreted saliva is unconvincing as rationale for this very large dose. The quantity of Francisella in organs after spread and growth is also not relevant. Rationale for injection of $10^{<8}$ extracellular vesicles as credibly representing the natural feeding is quite arbitrary. Quantities do matter in experimental design. In the

title "controlling virulence" is a stretch too far. It is not proven as it is based on comparison of only two tick-pathogen combinations. The intraperitoneal inoculation of LPS is a crude experiment with too many unknown components to interpret effectively. Conclusion extracellular vesicles are anti-inflammatory not an adequate characterization. Interpretation of splenomegaly, which is many mechanisms, as a measure of inflammation is an overinterpretation. Considering tick extracellular vesicles as a "molecular rheostat regulating virulence" is an overinterpretation of the data.
Conclusion: acceptance if the authors tone down their interpretations appropriately

RE: Response to Reviewers (Oliva Chavez et al., Nature Communications)

Note: Sections addressing the critiques of both reviewers are highlighted in yellow in the revised manuscript.

Reviewer #1 (Comments)

The authors have addressed many points which, in my opinion, were unclear to some degree in the first version of the manuscript. Nevertheless, some important information is still missing in the presentation of the proteomics data. Before going to the point, I would like to remark that this time I was able to access the proteomics data available on the PRIDE database. Whereas the Q-Exactive data looked ok to me (I was able to spot 4 proteins identified by a single peptide which were probably false positives, but on over 300 identifications, this is more or less within expected FDR), even though I could not explain why more than 40 proteins were DECOY hits, on a total of less than 400 proteins, Velos data were not of the same quality. This is of course due to the much higher sensitivity and mass accuracy of Q-Exactive HF. Nevertheless, Velos data are anyway reported in Table S2 despite the fact that protein identifications are sensibly less convincing. In my opinion, many identifications obtained in the first analysis are false (bad MS/MS data, very few peaks correctly assigned to b and y ions). It has to be stressed that the first analysis identified very few peptides, thus the FDR estimation algorithm might not have had sufficient data to perform correctly. Furthermore, it is significant to notice that half of the Velos identifications is not confirmed in the Q-Exactive analysis which, having identified 4x more proteins than Velos, should have identified the vast majority of proteins detected in the first analysis. This observation is also suggesting that many false identifications are present in the Velos data.

Due to the significant concerns from reviewer #1 regarding the identifications of proteins with the Velos instrument, we only presented the Q-Exactive analysis in the revised Table S2. The revised proteomics dataset has been deposited in the ProteomeXchange Consortium via the PRIDE partner repository with the current identifier PXD018779. This information can be accessed by Reviewer #1 following the information below:

Project Name: Label-Free Proteomics Profiling of Nanovesicles Isolated from Cultured Salivary Glands Isolated from Partially-Fed Adult Female *Ixodes scapularis*

Project accession: PXD018779

Project DOI: 10.6019/PXD018779

Reviewer account details:

Username: reviewer_pxd018779@ebi.ac.uk

Password: 939fiYrl

As the authors state, responding to my question number 2 (previous review):

"Additional proteins were exported from PEAKS with an FDR between 1.2% and 5%; these proteins were checked manually for their MS/MS (fragmentation profiles) and presence of unique peptides. Upon validation, these proteins were added to Table S2. From our experience using PEAKS, we observed that exporting proteins with FDR<5% helps improve the proteomic output, particularly in experimental condition, such as the one reported here, where the amount of material is scarce. However, to ensure correct peptide/protein assignment all the PEAKS imported data were re-evaluated with Scaffold and manually." The information on which protein was validated and added manually should be added to Table S2.

We included an asterisk to all proteins manually annotated in the revised Table S2. The MS/MS spectrum for all peptide hits identified with at least 1 unique peptide, together with the

tables for the identified ions were exported from PEAKS and are shown in the Supplementary Data 1. We included the criteria used for manual validation in the methods section of the manuscript (Page 29, Lines 627-637). A major issue in our mass spectrometry analysis was the technical difficulty for obtaining greater quantities of extracellular vesicles from the tick *I. scapularis*. These results are at the technical edge of what can be done in proteomics for non-model organisms, such as the tick *I. scapularis*.

The statement that many proteins were identified by manual validation is apparently not present in the manuscript. Please add this information to the Methods section.

This statement was included in the methods section (Page 29, Lines 627-637).

Besides, manual validation is a subjective procedure. The authors should present the validated and annotated MS/MS spectra as a separate file easily accessible in Supporting information in order to allow other researchers to assess the quality of the spectra. Needless to say, the MS/MS spectrum should strengthen the evidence for a poorly identified protein (just 1 peptide), thus it should be a spectrum which the majority of experts would consider of acceptable quality (unfortunately, this is also a subjective estimation). The criteria used for manual validation should be listed in the Methods section of the manuscript (presence of justifiable missed cleavages, abundant proline-directed fragments, series of at least 3 or at least 4 b or y ions, etc).

The MS/MS spectra for all protein hits identified with at least 1 unique peptide together with the tables for the identified ions were exported from PEAKS and are shown in the Supplementary Data 1. The criteria used for manual validation are listed in the methods section of the revised manuscript (Page 29, Lines 627-637). Briefly, all proteins identified by 1 unique MS/MS spectra and 1 unique peptide were further validated manually using the following criteria: 1) series of at least 3 or 4 b and y ions, altogether covering more than 80% of the identified peptide sequence; 2) correct assessment of fragments due to the neutral losses of H₂O (18 Da), carbonyl (CO) (28 Da) (*i.e.*, “a” ions for “b” series) and loss of phosphates (98 Da for H₃PO₄ and 80 Da for HPO₃); 3) correct assessment of internal fragments, mainly due to the presence of proline in the sequence. Proline residues normally fragment N-terminal to the residue, producing fragments in either direction; 4) correct assignments for the *m/z* and charge state of both the precursor and main “a”, “b” and “y” ions by inspecting the corresponding isotope envelopes.

Please find a list of observations which, in my opinion, need to be addressed in order to strengthen the presentation of proteomics data:

- Table S2. The following columns should be added: number of identified peptides, number of unique peptides, whether the protein was identified by manual validation of a MS/MS spectrum, whether the identification was from QExactive or Velos.

We revised the Supplementary Table S2. We added extra columns with the total number of identified peptides, the total number of unique peptides, the molecular weight of the identified proteins. We also denoted the peptides that were manually identified in our analysis with an asterisk in Table S2. The MS/MS spectra of these manually identified peptides were uploaded as supplemental information in our revised manuscript (Supplementary Data 1). As discussed above, we removed all data from Velos and presented only the Q-Exactive analysis. Additionally, we have added new tabs in Table S2 containing the original exported files from both PEAKS and Scaffold, including a new assessment of the PEAKS data set by including the total identified tryptic peptides containing posttranslational modifications (PTMs). All peptides and proteins have an identification score of $10\lg p > 20$ (corresponding to *p*-value < 0.01) using the FDR algorithm built-in in PEAKS.

- The Materials section should mention that manual validation was performed and should list the criteria adopted for manual validation.

The methods section has included all the manual validation criteria (Page 29, Lines 627-637).

- MS/MS spectra of manually validated identifications should be reported in Supporting information, together with annotation of b and y ions.

We provide the MS/MS spectra with the identified ions, as exported from the PEAKS software (Supplementary Data 1).

- Please update the correct PDX code in the manuscript (PXD013756)

The PDX code has been updated in the manuscript as PDX018779.

I do acknowledge that excluding 30-50 proteins from a list might have important consequences on downstream bioinformatics, causing a “chain reaction” of modifications which might have unpredictable consequences. Nevertheless, I do believe that, unfortunately, proteomics data present in this manuscript need to have a more conservative evaluation before being published.

Fortunately, no conceptual changes have occurred in the manuscript after revising the proteomics analysis.

Reviewer #2 (Comments)

The extensive responses to my queries and critiques address the issues that I have raised. Experimental design in this area is extremely difficult to achieve quantitative input that reflects natural tick feeding. The authors have done their best to justify their experimental designs.

No additional comments.

Reviewer #3 (Comments)

The following comments are addressed on behalf of Reviewer #3, who was unavailable to re-review. This manuscript represents a tremendous amount of high-quality experimental data. If the authors would tone down their interpretation and explain the weaknesses in their interpretations and data, the manuscript would merit publication in a journal of as high quality as *Nature Communications*.

We have changed the language of our revised manuscript so that our findings are interpreted in a more cautious manner. Whenever necessary, we also explained the technical constraints of working with ticks, microbes they transmit and extracellular vesicles.

This is a complex study involving advanced methods, e.g., tick salivary organoids and proteomics. These data are very broad and yielded only moderately overlapping results. Indeed, the study involved proteolytic analysis of tick salivary extracellular vesicles using two sources that yielded rather different results.

We have eliminated the data pertaining the Orbitrap Velos instrument. We are currently reporting only data acquired through the Q-Exactive mass spectrometer in the revised Table S2. The Q-Exactive mass spectrometer is a superior and more sensitive mass instrument (Sun *et al.*, 2013 - *Rapid Commun Mass Spectrom.*). A few proteins identified with the Q-Exactive platform in *I. scapularis* ticks were validated by western blot analysis (Figure 1h), strengthening our results. Additionally, we detected the salivary protein sialostatin L2 (SL2; B7PKZ1) in the western blot of extracellular vesicles from the saliva of tick *I. ricinus*, the arthropod vector of Lyme disease in Europe. These findings suggest that the identification of proteins in the cargo

of *I. scapularis* extracellular vesicles by Q-Exactive proteomics may also be used for closely related tick species. Altogether, the entire manuscript has been appropriately revised. The proteomics section is much improved when compared to earlier versions and the revised proteomics dataset has been deposited to the ProteomeXchange Consortium via the PRIDE partner repository with the current identifier PXD018779.

Study of the effect on dendritic epidermal cells was more focused. The authors need to address the drawbacks of the data.

In the revised manuscript, we have included a paragraph in the discussion section about the limitations of our findings (Page 17, Line 365 – Page 18, Line 376). For instance, we do not know whether the effect of tick extracellular vesicles on dendritic epidermal T cells (DETCs) is direct or indirect. The elongated dendrite morphology of DETCs enables contact with keratinocytes (Chodaczek *et al.* 2012 - *Nature Immunology*; Nielsen *et al.*, 2017 - *Nature Reviews Immunology*) and tick extracellular vesicles have been shown to affect wound healing in a human keratinocyte cell line (Zhou *et al.*, 2020 – *Frontiers in Cell and Developmental Biology*). Thus, future studies need to investigate whether tick extracellular vesicles affect keratinocytes in the epidermis. Investigators also need to ascertain the cellular composition of the epidermis in DETC-deficient (FVB-Taconic) and DETC-sufficient (FVB-Jackson) mice during a tick bite. FVB-Taconic, but not the FVB-Jackson mice, are depleted of DETCs because of an intrathymic differentiation defect due to a mutation in the *skint1* gene (Boyden *et al.* 2008 - *Nature Genetics*). Under normal conditions, DETCs interact with several neighboring cells in the epidermis. Conversely, the lack of DETCs in the epidermis results in increased keratinocyte death (Sharp *et al.* 2005 - *Nature Immunology*) and repopulation of the epidermis by $\alpha\beta$ T cells with a diverse T cell receptor repertoire (Nielsen *et al.*, 2017 - *Nature Reviews Immunology*).

The numerous references that were provided to address the critique of reviewer #2 included a general review with no data on ticks and data from unrelated tumor cells.

In the revised manuscript, we cited primary literature whenever possible. The lack of primary citations was due to the limited numbers of articles on tick extracellular vesicles.

The authors have failed to effectively address the hypothesis that extreme dilution of salivary gland extracellular vesicles after leaving the dermis would make systemic effects homeopathic.

In the revised manuscript, we emphasized that tick extracellular vesicles acted primarily on local skin immunity. We also have included sentences indicating that the effect of tick extracellular vesicles on systemic inflammation was minimal (Page 16, Lines 324-328). The systemic results observed with tick extracellular vesicles during *F. tularensis* infection were likely a consequence of molecular cues initiated at the skin site during a tick bite and distally propagated to other physiological systems.

Furthermore, the relevance of the 10^7 *Francisella tularensis* in the whole tick to its concentration in secreted saliva is unconvincing as a rationale for this very large dose. The quantity of *Francisella* in organs after spread and growth is also not relevant.

Unfortunately, there is no scientific consensus and universal approach that would broadly describe the *F. tularensis* concentration inside the tick and in the saliva during transmission. Each method has strengths and weaknesses and investigators have been debating this question for almost 100 years with no avail (Parker *et al.*, 1924 – Public Health Reports; Green, 1931 – *American Journal Epidemiology*). The crux of the matter is the use of different tick infection techniques, animal models, tick species, developmental stages and *Francisella* genotype/strains that lead to distinct research outcomes and obvious disagreements in the field.

Taking into consideration the constraints discussed above and to accommodate the request of reviewer #3, we rewrote this paragraph in the discussion section of the revised manuscript. We affirmed that our observations were only relevant to the animal model described in the manuscript. We elaborated that it remains unclear whether our findings could be extrapolated to infection occurring in the *bona fide* natural habitat of ticks (Page 16, Lines 329-336). We explained that our results describing the quantity of *F. tularensis* in organs after spread and growth should not be interpreted as a function of tick transmission. Rather, our findings should be construed as an outcome of disease in this animal model (Page 16, Lines 324-328). Collectively, we communicated to the reader the limitations of our findings.

Rationale for injection of 10^8 extracellular vesicles as credibly representing the natural feeding is quite arbitrary. Quantities do matter in experimental design.

Regrettably, it is technically unfeasible to determine the accurate number of extracellular vesicles secreted during tick salivation. The problem rests on the fact that feeding and salivation by ticks occur as an intermittent process for several days on a mammalian host (Biology of Ticks, 2014 – Daniel Sonenshine). This uneven activity changes the composition of molecules within the saliva (Kim *et al.*, 2016 - *PLoS Negl Trop Dis.*; Tirloni *et al.*, 2017 - *Front Cell Infect Microbiol.*). Therefore, precisely measuring the release of extracellular vesicles during tick salivation is not possible.

Additionally, there is not any technology that accurately measure the number of extracellular vesicles *in vivo*. Only a few laboratories around the world have developed fluorescent proteins that are tagged to the membrane to visualize extracellular vesicles by intravital microscopy or other sophisticated imaging techniques (Lai *et al.*, 2015 – *Nature Communications*; Verweij *et al.*, 2019 – *Developmental Cell*). These experimental approaches are not viable in ticks because there are not any tools for genetic manipulation of *I. scapularis* (de la Fuente, 2018 – *Ticks and Tick-Borne Diseases*).

Given these technical impediments, we determined a reasonable range of extracellular vesicles to use during experimentation. A review of the literature indicated various experimental designs with extracellular vesicles ranging from 10^6 to 10^8 molecules (Zamanian *et al.*, 2015 – *PLoS Neglected Tropical Diseases*; Szempruch *et al.*, 2016 – *Cell*; Sisquella *et al.*, 2017 – *Nature Communications*; Zhou *et al.*, 2018 – *PLoS Pathogens* and Vora *et al.*, 2018, *PNAS*). We then injected 1×10^8 extracellular vesicles in mice. In the revised manuscript, we acknowledged that it remains unclear whether our findings could be extrapolated to natural settings. Following the recommendation of reviewer #3, we discussed the limitations of our findings and explained our results in the context of the animal model developed in our manuscript (Page 16, Line 339 – Page 17, Line 351).

In the title "controlling virulence" is a stretch too far. It is not proven as it is based on comparison of only two tick-pathogen combinations.

We rewrote the title of our manuscript. It now reads: "***Tick Extracellular Vesicles Enable Arthropod Feeding and Promote Distinct Outcomes of Bacterial Infection***".

The intraperitoneal inoculation of LPS is a crude experiment with too many unknown components to interpret effectively.

We removed these experiments from the revised manuscript.

The conclusion that extracellular vesicles are anti-inflammatory not an adequate characterization.

We have removed these statements from the revised manuscript.

Interpretation of splenomegaly, which is many mechanisms, as a measure of inflammation is an overinterpretation.

We no longer use splenomegaly as a readout for inflammation in the revised text (Page 14, Lines 290-292).

Considering tick extracellular vesicles as a "molecular rheostat regulating virulence" is an overinterpretation of the data.

We have eliminated these statements from the manuscript.

Conclusion: acceptance if the authors tone down their interpretations appropriately.

We have interpreted our results more cautiously and discussed the limitations of our findings in the revised manuscript (see rewritten discussion section).

In closing, we are excited that the comments of both reviewers made our story much improved when compared to earlier versions of this manuscript. We thank *Nature Communications* for the manuscript evaluation.

Sincerely,

Joao Pedra, PhD
Associate Professor of Microbiology and Immunology

Reviewers' comments:

Reviewer #1 (Remarks to the Author):

All concerns have been fully addressed. The proteomics section has been improved. As a minor annotation, I wonder why the authors decided to add several variable modifications to their search parameters, in order to account for potential oxidative stress (p28, 600-611). Despite this broadening of the search parameters, the authors identify, as expected, mainly peptide deamidation and oxidation (usually on methionine) as variable modifications. In other words, the broadening of the search parameters has neither contributed to an increase in protein identification, nor to an unwanted increase in false positive rate.

Response to reviewer's comments are below:

REVIEWERS' COMMENTS

Reviewer #1 (Remarks to the Author):

All concerns have been fully addressed. The proteomics section has been improved. As a minor annotation, I wonder why the authors decided to add several variable modifications to their search parameters, in order to account for potential oxidative stress (p28, 600-611). Despite this broadening of the search parameters, the authors identify, as expected, mainly peptide deamidation and oxidation (usually on methionine) as variable modifications. In other words, the broadening of the search parameters has neither contributed to an increase in protein identification, not to an unwanted increase in false positive rate.

In any inflammatory condition, the presence of stress can generate oxidative moieties on proteins, which is why we have broadened our search parameters. In our experience, this approach improves the matching of PSM (peptides spectrum matches) to peptides, which leads to a higher sequence coverage, not necessarily followed by a change in the false discovery rate or leading to new protein IDs, as the reviewer pointed out.